# GTM: A General Time-series Model for Enhanced Representation Learning of Time-Series data

**Cheng He**[1,2], **Xu Huang**[1,2], **Gangwei Jiang**[1,2], **Zhaoyi Li**[1,2], **Defu Lian**[1,2]*,
**Hong Xie**[1,2]*, **Enhong Chen**[1,2], **Xijie Liang**[3], **Zengrong Zheng**[4], **Patrick P. C. Lee**[5]
[1] University of Science and Technology of China
[2] State Key Laboratory of Cognitive Intelligence    [3] Shanghai Black Wing Asset Management
[4] Di-Matrix Information Technology    [5] The Chinese University of Hong Kong
{cheng.he, xuhuangcs, gwjiang, lizhaoyi777}@mail.ustc.edu.cn
{liandefu, hongx87, cheneh}@ustc.edu.cn
lxjie@blackwingasset.com, zengrong.zheng@di-matrix.com
pclee@cse.cuhk.edu.hk

## Abstract

Despite recent progress in time-series foundation models, challenges persist in improving representation learning and adapting to diverse downstream tasks. We introduce a **G**eneral **T**ime-series **M**odel (**GTM**), which advances representation learning via a novel frequency-domain attention mechanism that captures time-granularity-aware features, an aspect underexplored in prior research. We further propose a novel pre-training strategy that unifies reconstruction and autoregressive objectives through a hybrid masking mechanism. Our pre-training strategy, combined with 2D positional encoding and span shuffling, enhances the robustness and generalization of representations. GTM is established as the first generative-task-agnostic model for time-series analysis, enabling seamless adaptation to various generative tasks without any task-specific modifications. Extensive experiments demonstrate that GTM consistently outperforms SOTA models on various generative tasks and achieves strong classification results with minimal adaptation. Furthermore, GTM exhibits clear scaling behavior, with accuracy improving as model size and pre-training data increase.

## 1 Introduction

**Foundation Models (FMs)** have achieved remarkable success, owing to their ability to learn rich representations from large-scale data and transfer effectively to diverse downstream tasks (Bommasani et al., 2021). However, extending these benefits to Time Series (TS) analysis remains challenging due to two major obstacles: (i) limited expressiveness of scalar, temporally indexed sequences, and (ii) wide heterogeneity of downstream tasks.

Recent advances in Time-Series Foundation Models (TSFMs) fall into two main categories: (i) **Forecasting-only FMs**, which are tailored for forecasting tasks and leverage temporal features such as lag covariates and adaptive patches (Rasul et al., 2023; Ekambaram et al., 2024; Shi et al., 2024); and (ii) **Multi-task FMs**, which employ autoregressive modeling, masked autoencoders, and contrastive learning to support multi-task adaptation (Liu et al., 2024b; Zhang et al., 2024; Dong et al., 2024; Goswami et al., 2024). While these models have improved feature extraction and generalization, they still require task-specific changes, especially for generative tasks, and rarely explore new perspectives beyond typical time-domain features.

In multi-task TS analysis, downstream tasks are generally categorized as either **generative** (e.g., forecasting, imputation, anomaly detection), which requires modeling the underlying data distribution, or **discriminative** (e.g., classification), which maps TS inputs to categorical labels. Although recent

---

*Corresponding authors

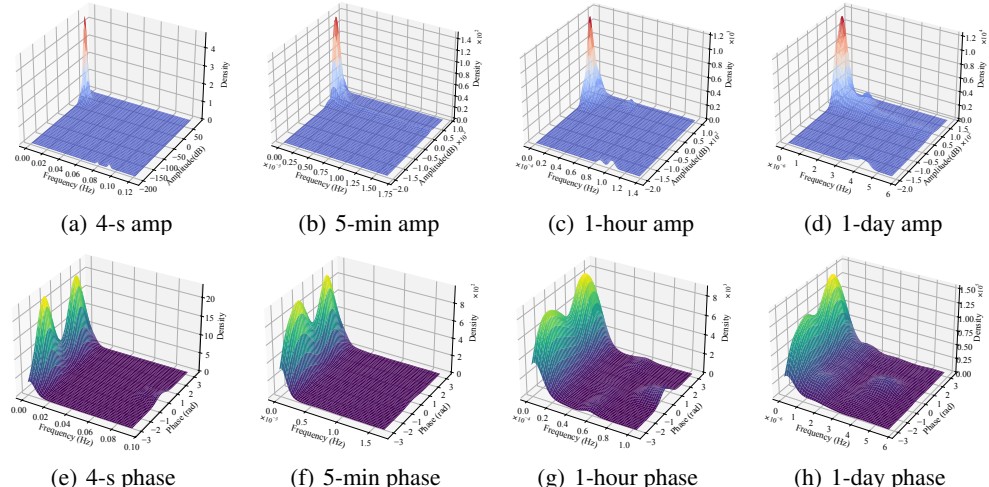

|  |  |  |  |
|---|---|---|---|
| (a) 4-s amp | (b) 5-min amp | (c) 1-hour amp | (d) 1-day amp |
| (e) 4-s phase | (f) 5-min phase | (g) 1-hour phase | (h) 1-day phase |

Figure 1: Amplitude- and phase-frequency joint distributions for TS data with varying granularities.

TSFMs can handle multiple generative tasks (Liu et al., 2024b; Zhang et al., 2024) or adapt across both categories of tasks (Dong et al., 2024; Gao et al., 2024), they typically require modifications at the token, pre-training, or projection-header levels to achieve such flexibility. To our knowledge, no TSFM can adapt to all generative tasks in a truly task-agnostic manner without such changes.

In this work, we present a comprehensive analysis of large-scale, multi-domain TS data using Fast Fourier Transform and 2D Kernel Density Estimation to estimate the joint probability distributions of amplitude-frequency and phase-frequency at various temporal granularities. As shown in Figure 1, these distributions differ significantly across time granularities, highlighting a critical but unexplored dimension in TS representation learning. This empirical observation directly informs our model design, motivating the development of frequency-domain network modules tailored to capture such multi-granularity representations.

Building on these insights, we propose a **General Time Series Model (GTM)**, which explicitly incorporates time granularity as a key factor for robust TS representation. To enable effective adaptation to generative tasks, we introduce a novel pre-training framework that unifies reconstruction and autoregressive objectives via a hybrid masking strategy. Our framework combines random and controlled consecutive tail masking, 2D positional encoding, and span shuffling. This design enables GTM to learn robust and generalizable representations, allowing seamless adaptation to a wide range of generative tasks without any task-specific modifications.

**Our main contributions are:**

- We design GTM, a TSFM built with a novel Fourier attention mechanism to capture distributional differences across temporal granularities, substantially improving TS representation quality.

- We propose a unified pre-training framework that integrates hybrid masking, 2D positional encoding, and span shuffling, jointly optimizing reconstruction and autoregressive objectives to enhance robustness and generalizability. This establishes GTM as the first generative-task-agnostic TSFM.

- Extensive experiments demonstrate that GTM consistently outperforms state-of-the-art baselines across a variety of benchmarks, offering scalable and cost-effective performance suitable for industrial applications.

The code and implementation details are available at: `https://github.com/MMTS4All/GTM`.

## 2 RELATED WORK

We focus on TSFMs trained from scratch. Additional literature survey can be found in Section B.1.

**Early Attempts.** Early FMs adapted techniques for TS tasks, thereby laying the foundation for TSFMs. For example, TimesNet (Wu et al., 2023) transforms 1D time series into 2D feature maps

using CNNs to capture multi-periodicity patterns, while adding task-specific projection headers for diverse generative tasks. Similarly, PatchTST (Nie et al., 2023) enhances pre-trained Transformers for forecasting by learning channel-independent, inter-patch representations. Despite their progress, these models fall short of TSFM standards due to the lack of large-scale pre-training and effective adaptation across diverse tasks.

**TSFMs for Forecasting.** Lag-Llama (Rasul et al., 2023) and GPHT (Liu et al., 2024c) both utilize decoder-only architectures to model temporal dependencies, with Lag-Llama incorporating lagged covariates and timestamp features, while GPHT employs a hierarchical backbone for long-term forecasting across arbitrary time horizons. TimesFM (Das et al., 2024) pushes the boundaries by using a stacked Transformer pretrained on $O(100B)$ data points to learn domain-invariant representations. Other works, such as GPD (Yang et al., 2024) and UTSD (Ma et al., 2024), explore the use of diffusion models to capture cross-domain correlations and improve robustness across diverse forecasting tasks. MOIRAI (Woo et al., 2024) and TTMs (Ekambaram et al., 2024) focus on multivariate time series forecasting, with MOIRAI tackling cross-frequency learning through a masked Transformer architecture and TTMs emphasizing the learning of cross-channel correlations. Finally, TIME-MOE (Shi et al., 2024) introduces a MOE design that offers flexibility and supports multi-resolution forecasting. Despite these advancements, most models primarily focus on modeling temporal dependencies and do not fully exploit richer, multi-domain information (e.g., frequency-domain features) that could enhance the ability to address more complex forecasting tasks.

**Multi-task TSFMs.** Recent work has greatly advanced the adaptability of TSFMs for diverse tasks. UP2ME (Zhang et al., 2024) combines Masked AutoEncoder pretraining with Graph Transformer fine-tuning for flexible adaptation. Timer (Liu et al., 2024b) adopts an autoregressive, causal-attention framework, pretraining on unified sequences to improve generalization. For discriminative tasks, TimeSiam (Dong et al., 2024) applies Siamese contrastive learning, while LPTM (Kamarthi & Prakash, 2023) fuses Transformer and GRU modules to extract robust tokenized representations from heterogeneous data. UniTS (Gao et al., 2024) introduces task tokenization within a dual-tower Transformer, supporting both generative and classification tasks. Overall, masked reconstruction and contrastive learning are oriented towards representation learning by capturing intra-sequence patterns and inter-sequence similarities, respectively, with downstream adaptation typically achieved by replacing the projection head. Predictive pretraining, on the other hand, focuses on modeling long-term temporal dependencies to forecast multi-step future outcomes, making it particularly suited to downstream predictive tasks. However, due to the absence of a unified pretraining objective, these models require task-specific modifications at the tokenization (e.g., UniTS), pre-training strategy (e.g., Timer), or model level (e.g., UP2ME, TimeSiam, LPTM) to achieve strong downstream performance.

## 3 METHOD

### 3.1 DESIGN OVERVIEW

We denote a TS by $\boldsymbol{X} = [X_{c,t} : c \in [C], t \in [T]]$, where $C$ and $T$ are the number of variables and timestamps, respectively. We pre-train our model, GTM, from scratch on the large-scale UTSD-12G dataset (Liu et al., 2024b), which covers diverse application domains. Figure 2 shows the overall architecture:

- **Input Embedding:** We apply Reversible Instance Normalization (Kim et al., 2022), Channel Independence (CI), patching (Nie et al., 2023), and masking (Du et al., 2022) to transform raw TS data into univariate masked token sequences. Each token is further enriched with linear and positional embeddings before entering the backbone.

- **N-stack Decoder-only Backbone:** GTM uses a decoder-only Transformer backbone to generate outputs autoregressively. To capture both temporal and frequency-domain information, we retain a temporal self-attention module and design the Fourier attention module (details in Section 3.2).

- **Output Projection:** A unified linear projection layer, followed by instance denormalization, produces outputs autoregressively for both pretraining and downstream tasks.

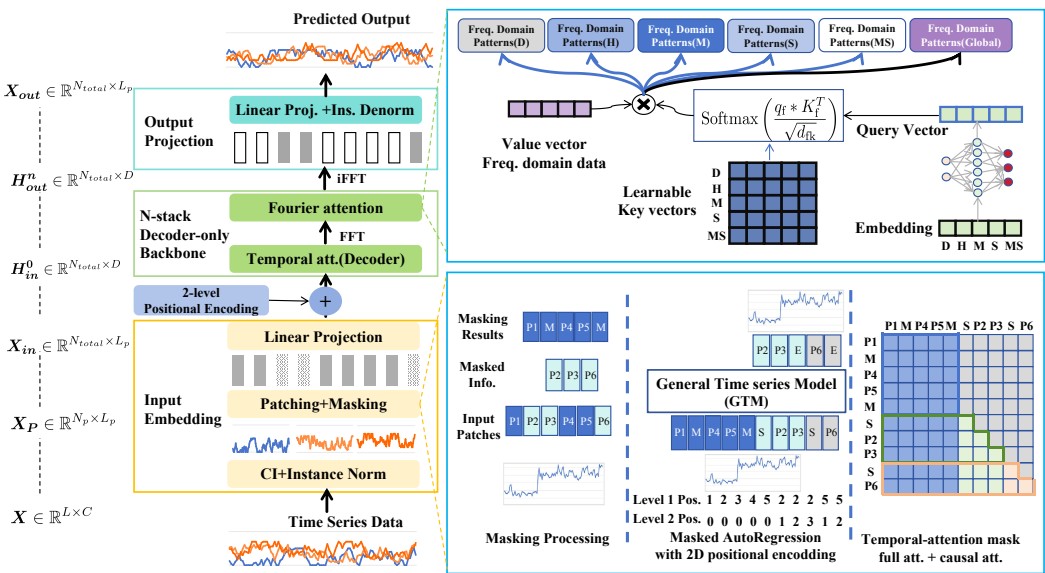

Figure 2: GTM model architecture for pre-training. **Left**: TS data pass through three key components: input embedding, $N$-stack Transformer backbone, and output projection, to generate reconstruction results autoregressively. **Lower right**: Patching and masking using both full and causal attention mechanisms, adapted from the NLP field and optimized for TS pre-training. **Upper right**: A Fourier attention module designed to learn representation of TS data with varying granularities. Pseudo-code of GTM architecture and pre-training strategy is provided in Algorithm 1 in Appendix B.2.3.

## 3.2 N-STACK DECODER-ONLY BACKBONE

We design an $N$-stack decoder-only backbone that jointly models temporal and frequency patterns in TS data. Each decoder block consists of a standard temporal self-attention layer followed by a Fourier attention module, which incorporates frequency-domain information via FFT. To enable granularity-aware frequency modeling, we represent time granularity as a quintuple: (day, hour, minute, second, millisecond). For example, the ETTm dataset (Wu et al., 2021) is encoded as [0, 0, 15, 0, 0]. We also introduce five learnable key embeddings, each for a typical granularity. Attention weights are computed by taking the dot product of the query with each key, followed by softmax normalization, and used to combine five corresponding frequency learning matrices. In addition, a global frequency learning module operates in parallel to capture patterns not tied to any specific time granularity. This module is always active and complements the granularity-specific modules.

**Temporal & Fourier Attention.** Given the embedded input $\boldsymbol{H}_{in} \in \mathbb{R}^{N_{total} \times D}$—where $N_{total}$ is the total number of masked and reconstructed patches and $D$ is the embedding dimension—the temporal self-attention module computes

$$\boldsymbol{H}_{\text{TemAttOut}} = \texttt{Self\_Attention}(\boldsymbol{Q}_h, \boldsymbol{K}_h, \boldsymbol{V}_h) \in \mathbb{R}^{N_{total} \times D}, \tag{1}$$

where $\boldsymbol{Q}_h = \boldsymbol{H}_{in}\boldsymbol{W}_h^Q$, $\boldsymbol{K}_h = \boldsymbol{H}_{in}\boldsymbol{W}_h^K$, and $\boldsymbol{V}_h = \boldsymbol{H}_{in}\boldsymbol{W}_h^V$ are linear projections with learnable weight matrices. Next, a column-wise FFT transforms each temporal patch into the frequency domain:

$$\boldsymbol{H}_{\text{FFT}} = \texttt{FFT}(\boldsymbol{H}_{\text{TemAttOut}}). \tag{2}$$

To capture frequency-specific patterns, we design six frequency-domain modules: five low-rank modules for five granularities, parameterized by $\{\boldsymbol{A}_i, \boldsymbol{B}_i\}_{i=1}^5$, and one global module with full connection $\boldsymbol{W}_{\text{full}}$. The time granularity is encoded as a quintuple and embedded into a query vector $\boldsymbol{q}_f = \boldsymbol{q}\boldsymbol{W}_f^Q$. Five learnable key vectors $\boldsymbol{K}_f$ represent the corresponding granularities. Fourier attention weights are computed as $\alpha = \texttt{SoftMax}\left(\frac{\boldsymbol{q}_f \boldsymbol{K}_f^\top}{\sqrt{d_{fk}}}\right)$ and are subsequently used to aggregate the outputs of the five low-rank modules:

$$\boldsymbol{H}_{\text{FourierAtt}} = \sum_{i=1}^5 \alpha_i (\boldsymbol{A}_i \boldsymbol{B}_i) \boldsymbol{H}_{\text{FFT}} + \boldsymbol{W}_{\text{full}} \boldsymbol{H}_{\text{FFT}}. \tag{3}$$

The final output is obtained by applying the inverse FFT:

$$\boldsymbol{H}_{\text{out}} = \texttt{iFFT}(\boldsymbol{H}_{\text{FourierAtt}}) \in \mathbb{R}^{N_{total} \times D}. \tag{4}$$

This process is repeated for $N$ stacked decoder-only layers, with each layer taking the output of the previous layer as input:

$$\boldsymbol{H}_{out}^{(n)} = \texttt{GTM\_Decoder}(\boldsymbol{H}_{in}^{(n)}), \quad \boldsymbol{H}_{in}^{(n)} = \boldsymbol{H}_{out}^{(n-1)}, \tag{5}$$

where $n \in [N]$ and $\boldsymbol{H}_{in}^{(1)} = \boldsymbol{H}_{in}$.

**Output Projection:** A unified linear projection maps the backbone output to patch-level predictions:

$$\boldsymbol{X}_{out} = \boldsymbol{W}_{\text{LinProj}} \cdot \boldsymbol{H}_{out}^{(N)}. \tag{6}$$

This enables GTM to support various generative tasks without further architectural changes.

### 3.3 PRE-TRAINING FRAMEWORK

We divide each time series into overlapping patches using CI and patching (Nie et al., 2023). For each variable, the series is split into overlapping windows of length $L$ and stride $\tau$, as $\boldsymbol{X}_i = [X_{c,i \times \tau}, \dots, X_{c,i \times \tau + L - 1}]$, then divided into $N_p$ patches. Inspired by GLM (Du et al., 2022), we use a hybrid masking strategy:

- Randomly sample $\ell$ patch spans (each a consecutive group of patches).
- Randomly permute the sampled spans, and pad learnable vectors **[START]** and **[END]** tokens to form input and target sequences.
- Replace each span with a single **[MASK]** token to create a corrupted input.
- Apply a controlled proportion of consecutive **[MASK]** tokens to at the tail.

Specifically, we introduce a hyperparameter $pred\_ratio$ to flexibly control the probability of applying consecutive tail masking. As an example, for each training instance, a random variable $r \sim \mathcal{U}(0, 1)$ is sampled and a corrupted input can be constructed as follows:

$$\mathbf{X}_{\boldsymbol{P}crpt} = \begin{cases} [\mathbf{X}_1, \dots, \mathbf{X}_{N_p - k}, \underbrace{\texttt{[MASK]}, \dots, \texttt{[MASK]}}_{k}], & \text{if } r \leq \texttt{pred\_ratio} \\ \text{RandomMask}(\mathbf{X}_P), & \text{otherwise} \end{cases}$$

where $k = \lfloor \alpha N_p \rfloor$, and $\alpha$ representing the tail masking ratio. This approach unifies mask reconstruction and autoregressive forecasting under a single pre-training objective, enabling the model to learn both general representations and future prediction capabilities. Based on this strategy, we can get:

$$\boldsymbol{X}_{\text{in}} = [\boldsymbol{X}_{\boldsymbol{P}\text{crpt}}, [S], \boldsymbol{S}_{\sigma(1)}, \dots, [S], \boldsymbol{S}_{\sigma(\ell)}] \tag{7}$$

$$\boldsymbol{Y} = [\boldsymbol{S}_{\sigma(1)}, [E], \dots, \boldsymbol{S}_{\sigma(\ell)}, [E]] \tag{8}$$

where $\boldsymbol{X}_{\boldsymbol{P}\text{crpt}}$ denotes the masked input, $\sigma(\cdot)$ is a random permutation. The pre-training objective is to autoregressively reconstruct all masked patches by minimizing MSE:

$$\mathbb{P}(\boldsymbol{X}_{out}) = \prod_i \mathbb{P}(\boldsymbol{X}_{\boldsymbol{out}i} | \boldsymbol{X}_{\boldsymbol{P}\text{crpt}}, \boldsymbol{S}_{\sigma(j \leq i)}) \tag{9}$$

$$\text{Loss}_{MSE} = \frac{1}{|\boldsymbol{Y}|} \sum_i \|\boldsymbol{X}_{\boldsymbol{out}i} - \boldsymbol{y}_i\|^2 \tag{10}$$

Before feeding to the backbone, we apply trainable linear embedding and 2D positional encoding (Du et al., 2022), ensure that the backbone model is aware of the length of the masked span when generating output patches:

$$\boldsymbol{H}_{in} = \boldsymbol{W}_{emb}\boldsymbol{X}_{in} + \boldsymbol{W}_{1D\_pos} + \boldsymbol{W}_{2D\_pos} \tag{11}$$

We employ full attention for masked reconstruction and causal attention for autoregressive generation, effectively preventing information leakage.

Table 1: Avg. MSE & MAE forecasting results. Results are averaged over varying prediction lengths. **Bold** & underline indicate the best & 2nd-best results respectively. See full results in Table 18.

| Models | GTM | | GPT4TS | | UniTS-PMT | | TTM_E | | PatchTST | | TimesNet | | DLinear | | FEDformer | | Autoformer | | Informer | |
|---|---|---|---|---|---|---|---|---|---|---|---|---|---|---|---|---|---|---|---|---|
| dataset | MSE | MAE | MSE | MAE | MSE | MAE | MSE | MAE | MSE | MAE | MSE | MAE | MSE | MAE | MSE | MAE | MSE | MAE | MSE | MAE |
| ETTh1 | .404 | .429 | .427 | **.426** | .461 | .454 | **.402** | - | .413 | .434 | .458 | .450 | .422 | .437 | .428 | .453 | .473 | .476 | 1.040 | .795 |
| ETTm1 | **.339** | **.376** | .352 | .383 | - | - | .350 | - | .352 | .382 | .400 | .406 | .357 | .378 | .382 | .422 | .515 | .493 | .961 | .734 |
| Weather | **.225** | .266 | .237 | .270 | .243 | .273 | .234 | - | .225 | **.263** | .259 | .287 | .246 | .300 | .332 | .375 | .335 | .379 | .634 | .548 |
| Traffic | **.385** | .266 | .414 | .294 | .494 | .313 | .385 | - | .390 | **.263** | .620 | .336 | .433 | .295 | .603 | .372 | .616 | .383 | .764 | .416 |
| Electricity | .161 | .254 | .167 | .263 | .184 | .282 | **.158** | - | .159 | **.252** | .192 | .295 | .166 | .263 | .207 | .321 | .214 | .326 | .311 | .397 |

## 3.4 FINE-TUNING FOR DOWNSTREAM TASKS

Due to its unified architecture and pre-training strategy, GTM yields robust representations and supports a wide range of generative downstream tasks without task-specific modifications, aside from minor preprocessing (e.g., removing masking and 2D positional encoding). This versatility enables GTM to deliver high-precision results across diverse time series applications (Section 4).

## 4 EXPERIMENTS

We extensively evaluate GTM primarily on generative tasks, while also including discriminative tasks, to demonstrate its advanced representation learning and seamless multi-task adaptability. Across all tasks, GTM is compared with state-of-the-art baselines (see Appendix B.2.2). We further analyze the benefits of large-scale pre-training, generalization in zero-shot and few-shot settings, and perform ablation and scalability studies. Finally, we assess the computational overhead of GTM's key components and its overall model efficiency. Additional results on hyperparameter sensitivity analysis are provided in Appendix B.3.7 and B.3.8, confirming GTM's cost-effectiveness and industrial applicability.

### 4.1 DATASETS DESCRIPTION

We use the large-scale public TS dataset, UTSD-12G, for pre-training, ensuring no downstream task-related data is included to prevent leakage. We conduct experiments on five widely used public datasets for forecasting and imputation (Wu et al., 2021), five popular labeled datasets for anomaly detection (Su et al., 2019; Hundman et al., 2018; Mathur & Tippenhauer, 2016; Abdulaal et al., 2021), and ten standard datasets for classification (Bagnall et al., 2018). The detailed statistics of these public datasets are provided in Appendix B.2.1.

### 4.2 LONG-TERM FORECASTING

For long-term forecasting, we select representative baselines and cite their results respectively. These SOTA models include the LLM-enhanced model GPT4TS (Zhou et al., 2023), the multi-task TSFM UniTS-PMT (Gao et al., 2024), the task-specific TSFM $TTM_E$, TimesNet (Ekambaram et al., 2024; Wu et al., 2023), the Transformer-based models PatchTST, FEDformer, Autoformer, Informer (Nie et al., 2023; Zhou et al., 2022; Wu et al., 2021; Zhou et al., 2021), and the MLP-based model Dlinear (Zeng et al., 2023). We focus on baselines that align closely with our experimental settings, excluding models that require pre-training and fine-tuning on the same datasets for downstream tasks. The long-term forecasting lengths include $T \in \{96, 192, 336, 720\}$ time points. We use MSE and MAE as evaluation metrics. Notably, GTM directly utilizes a pre-trained model without any modifications. As shown in Table 1, GTM outperforms all SOTA models, achieving the highest total number of best- and 2nd-best-place results across tests with varying forecasting lengths, while PatchTST ranks second. Full results, additional baseline comparisons with SOTA TSFMs Sundial and Time-MOE, and error bar analysis with 95% confidence intervals and more experiments on extended challenging, real-world datasets are provided in Appendix B.3.1.

Table 2: Avg. MSE & MAE results of Imputation. Results are averaged over varying data missing ratios at the time-point level. **Bold** and underline denote the best and the 2nd-best results, respectively. Full results are listed in Table 22.

| Models | GTM | | GPT4TS | | TimesNet | | PatchTST | | DLinear | | Fedformer | | Informer | |
|---|---|---|---|---|---|---|---|---|---|---|---|---|---|---|
| Dataset | MSE | MAE | MSE | MAE | MSE | MAE | MSE | MAE | MSE | MAE | MSE | MAE | MSE | MAE |
| ETTh1 | **0.053** | **0.152** | 0.069 | 0.173 | 0.078 | 0.187 | 0.115 | 0.224 | 0.201 | 0.306 | 0.117 | 0.246 | 0.161 | 0.279 |
| ETTm1 | **0.021** | **0.096** | 0.028 | 0.105 | 0.027 | 0.107 | 0.047 | 0.140 | 0.093 | 0.206 | 0.062 | 0.177 | 0.071 | 0.188 |
| weather | **0.030** | **0.054** | 0.031 | 0.056 | 0.030 | 0.054 | 0.060 | 0.144 | 0.052 | 0.110 | 0.099 | 0.203 | 0.045 | 0.104 |
| Electricity | 0.086 | 0.202 | 0.090 | 0.207 | 0.092 | 0.210 | **0.072** | **0.183** | 0.132 | 0.260 | 0.130 | 0.259 | 0.222 | 0.328 |

Table 3: The F1 scores for the anomaly detection tasks.

| Models | GTM | UP2ME | GPT4TS | TimesNet | PatchTST | FEDformer | DLinear | Autoformer | Informer |
|---|---|---|---|---|---|---|---|---|---|
| Dataset | F1(%) | F1(%) | F1(%) | F1(%) | F1(%) | F1(%) | F1(%) | F1(%) | F1(%) |
| MSL | 82.53 | - | 82.45 | 81.84 | 78.70 | 78.57 | **84.88** | 79.05 | 84.06 |
| SMAP | **77.57** | - | 72.88 | 69.39 | 68.82 | 70.76 | 69.26 | 71.12 | 69.92 |
| SWaT | **94.78** | 93.85 | 94.23 | 93.02 | 85.72 | 93.19 | 87.52 | 92.74 | 81.43 |
| SMD | 85.47 | 83.31 | **86.89** | 84.61 | 84.62 | 85.08 | 77.10 | 85.11 | 81.65 |
| PSM | 95.43 | 97.16 | 97.13 | **97.34** | 96.08 | 97.23 | 93.55 | 93.29 | 77.10 |
| Average | **87.01** | - | 86.72 | 85.24 | 82.79 | 84.97 | 82.46 | 84.26 | 78.83 |

## 4.3 IMPUTATION

We use the same publicly available datasets in forecasting tasks and follow the protocol proposed by (Zhou et al., 2023) for imputation tasks. To align with benchmark settings, we apply point-wise missing ratios for interpolation, and directly use pre-trained model for fine-tuning, only omitting the patching process. The point-wise imputation baselines include GPT4TS, TimesNet, PatchTST, FEDformer, Informer, and Dlinear. We conduct the task with varying missing data ratios of $\{12.5\%, 25\%, 37.5\%, 50\%\}$ at the time-point level. Table 2 demonstrates that, even without patch preprocessing, GTM achieves significant performance improvements. Compared to the second-best model, GTM gets a 23.1% reduction in MSE, 12.1% in MAE for ETTh1 data, and 25.0% reduction in MSE, 8.6% in MAE for ETTm1 data. More details are in Appendix B.3.2

## 4.4 ANOMALY DETECTION

For anomaly detection, we fine-tune the pre-trained GTM model in a self-supervised manner via data reconstruction, without any task-specific modifications. Following a standard approach (Xu et al., 2018), points with reconstruction errors above a threshold are labeled as anomalies. We compare GTM against baselines, including the multi-task TSFMs (UP2ME, TimesNet), the LLM-enhanced model (GPT4TS), transformer-based models (PatchTST, FEDformer, Autoformer, Informer), and the MLP-based model (DLinear). As shown in Table 3, GTM achieves the highest F1 score across all baselines, with improvements ranging from 0.33% (over GPT4TS) to 10.38% (over Informer). We also report results on the TSB-AD datasets, using various widely used measures (Liu & Paparrizos, 2024), along with a broad coverage of TSFMs' test results. See Appendix B.3.3 for details.

## 4.5 CLASSIFICATION

Although GTM is designed as a generative-task-agnostic foundation model, it can be extended to discriminative tasks like classification. As outlined in Section 1, we adapt only the output projection layer to map inputs to categorical labels, leaving the rest of the model unchanged. Following this approach, we fine-tune pre-trained GTM on 10 widely-used classification datasets (Bagnall et al., 2018), with accuracy as the evaluation metric. As shown in Table 4, GTM achieves the most best-(5) and second-best(4) results compared to SOTA multi-task TSFMs.

Table 4: The Accuracy results of Classification tasks.

| Dataset/Model | GTM | UNITS-SUP | UNITS-PMT | GPT4TS | TimesNet | iTransformer |
|---|---|---|---|---|---|---|
| EthanolConcentration | 34.2 | / | / | 34.2 | **35.7** | 28.1 |
| FaceDetection | **69.9** | 65.4 | 58 | 69.2 | 68.6 | 66.3 |
| Handwriting | **34.8** | / | / | 32.7 | 32.1 | 24.2 |
| Heartbeat | 77.5 | 63.9 | 65.4 | 77.2 | **78** | 75.6 |
| Japanese Vowels | 92.1 | 92.2 | 90.3 | **98.6** | 98.4 | 96.6 |
| PEMS-SF | 88.4 | 83.2 | 82.7 | 87.9 | **89.6** | 87.9 |
| SelfRegulationSCP1 | 92.5 | / | / | **93.2** | 91.8 | 90.2 |
| SelfRegulationSCP2 | 60 | 48.9 | 57.2 | 59.4 | 57.2 | 54.4 |
| SpokenArabicDigits | **99.2** | 96.8 | 95.5 | 99.2 | 99 | 96 |
| UWaveGestureLibrary | **89.3** | 82.2 | 85.3 | 88.1 | 85.3 | 85.9 |
| Best Count | **5** | 0 | 0 | 2 | 3 | 0 |

Table 5: Zero-shot capability (MSE) of GTM compared to SOTA TSFMs.

| Dataset | GTM | TIMER-1B | MOIRAI-S | MOMENT | TimesFM | CHRONOS-S1 |
|---|---|---|---|---|---|---|
| ETTh1 | **0.407** | 0.438 | 0.441 | 0.674 | 0.414 | 0.571 |
| ETTm1 | 0.593 | 0.690 | 0.562 | 0.670 | **0.354** | 0.632 |
| Weather | **0.172** | 0.181 | 0.195 | 0.255 | - | - |
| Electricity | **0.187** | 0.192 | 0.212 | 0.744 | - | - |
| Traffic | 0.542 | **0.458** | 0.616 | 1.293 | - | - |
| Average | **0.380** | 0.392 | 0.405 | 0.727 | - | - |

## 4.6 EFFECTIVENESS OF PRE-TRAINING

By pre-training on large-scale TS data across multiple temporal granularities, GTM learns richer, more diverse patterns. We first demonstrate the effectiveness of pre-training by evaluating GTM's generalization in zero-shot and few-shot settings. Table 5 shows that, compared to 5 SOTA TSFMs: Timer, MOIRAI-S(Woo et al., 2024), MOMENT(Goswami et al., 2024), TimesFM(Das et al., 2024), and Chronos-S1(Ansari et al., 2024), GTM ranks first on average MSE across 5 datasets with a forecasting length of 96 in zero-shot. In few-shot testing, Figure 3 shows that GTM outperforms TimesFM across 4 forecasting lengths on ETTm1 data, achieving better performance with only 10% of the data for fine-tuning, improving results with the largest MSE reduction of 7.53%.

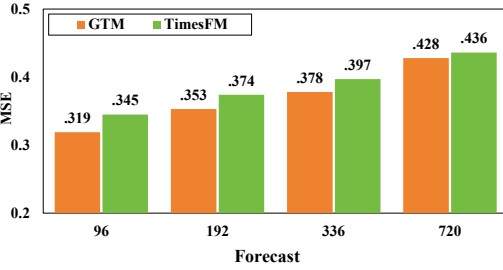
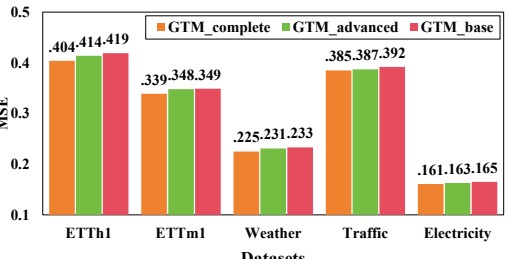

Figure 3: GTM VS. TimesFM in few-shot on ETTm1 dataset, 10% samples for fine-tuning.

Figure 4: Average results of long-term forecasting in ablation test.

We also compare the fine-tuned GTM pre-trained on UTSD datasets with the baseline GTM, which is trained directly on task-specific datasets with random initialization. This further highlights the benefits of pre-training across various tasks. Tables 6 and 7 present the average performance of both models across all datasets, covering forecasting tasks with varying prediction lengths and imputation tasks with different missing data ratios. The results show that, for forecasting, the fine-tuned GTM consistently outperforms the baseline GTM in every comparison. It achieves reductions in MSE of 0.5%-7.8% and in MAE of 0.8%-8.0%. Similarly, for imputation, the fine-tuned GTM also outperforms the baseline GTM, achieving MSE reductions of 1.2%-11.7% and MAE reductions of 0.5%-14.2%. More details are provided in Appendix B.3.4. For anomaly detection, Table 8 shows that with pre-training, the fine-tuned GTM model achieves performance gains across all test datasets, with an average increase of 1.2% in F1-score compared to the baseline GTM model.

Table 6: Avg. results of forecasting results compared with GTM model w/o pre-train. Table 24 shows full results in Appendix B.3.4.

| Models | GTM | | GTM no pretrain | |
|---|---|---|---|---|
| dataset | MSE | MAE | MSE | MAE |
| ETTh1 | **0.404** | **0.429** | 0.435 | 0.447 |
| ETTm1 | **0.339** | **0.376** | 0.351 | 0.389 |
| Weather | **0.225** | **0.266** | 0.244 | 0.289 |
| Traffic | **0.385** | **0.266** | 0.387 | 0.268 |
| Electricity | **0.161** | **0.254** | 0.163 | 0.256 |

Table 7: Avg. Imputation results compared with GTM model without pre-training. Table25 in Appendix B.3.4 shows the full results.

| Models | GTM | | GTM no pretrain | |
|---|---|---|---|---|
| dataset | MSE | MAE | MSE | MAE |
| ETTh1 | **0.053** | **0.152** | 0.055 | 0.156 |
| ETTm1 | **0.021** | **0.096** | 0.023 | 0.100 |
| Weather | **0.030** | **0.054** | 0.034 | 0.063 |
| Electricity | **0.086** | **0.202** | 0.087 | 0.203 |

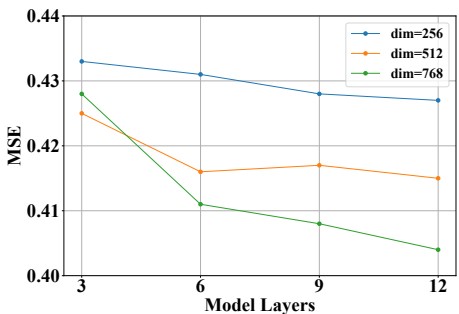

Figure 5: Model scalability analysis (ETTh1).

Table 8: Anomaly detection results compared with GTM model without pre-training.

| Models | GTM | GTM no pretrain |
|---|---|---|
| dataset | F1(%) | F1(%) |
| MSL | **82.53** | 81.92 |
| SMAP | **77.57** | 76.48 |
| SWaT | **94.78** | 94.66 |
| SMD | **85.47** | 82.11 |
| PSM | **95.43** | 95.42 |
| Average | **87.15(+1.2%)** | 86.11 |

## 4.7 ABLATION TESTS

We conduct ablation experiments on long-term forecasting tasks across different prediction lengths to evaluate the key components in the GTM model. We use a baseline version of the GTM model without the frequency-domain analysis module and compare it with an advanced version that lacks the time-granularity-aware modules. By also comparing both with the complete GTM model, we gain insights into the impact of these key design elements.

Figure 4 shows the average forecasting results for each dataset. The complete GTM model outperforms all other models in every test. The advanced GTM model ranks second. This demonstrates that combining temporal and frequency-domain analysis, especially time granularity-aware modules, enables the GTM model to effectively learn distribution representations from TS datasets with varying time granularities. More details of ablation tests are given in Appendix B.3.5.

## 4.8 SCALABILITY ANALYSIS

FMs generally adhere to scaling laws, where their accuracy and capabilities scale predictably with both model size and training data (Kaplan et al., 2020). This is crucial for FM design and deployment. To explore the scalability of GTM, we pre-train the model with increasing model size (layers and dimensions) and data size, and conduct forecasting tests on various downstream tasks. Figure 5 shows the average forecasting results on the ETTh1 dataset for various forecasting lengths, including $T \in \{96, 192, 336, 720\}$ time points, using pre-trained models with different numbers of layers and embedding dimensions. The results indicate that GTM follows scaling laws, achieving a better MSE with deeper and wider models. However, when the model depth is insufficient, increasing the width (embedding dimension) may not improve performance.

We also pre-trained GTM on different scales of the UTSD dataset and evaluated its forecasting performance for various forecasting lengths on the ETTh1 and Weather datasets with fine-tuning. Figure 6 in Appendix B.3.6 shows that GTM performs better with larger pre-training datasets, as evidenced by the average MSE results, consistent with the expected data scaling laws.

## 4.9 COMPUTATIONAL OVERHEAD AND EFFICIENCY ANALYSIS

We compare GTM with four reproduced baseline models, including three TSFMs: Time-MOE(base), GPT-2(6)-768, TimesNet-768, and one deep learning model: FEDformer-768, in terms of model size

Table 9: Comparison of model parameters and efficiency. Training and inference speeds are reported in seconds per iteration. An iteration is a full forward-backward pass over a batch for training and a forward pass for inference. All timings exclude data loading and logging overheads.

| Model | Parameter | Training Speed | Inference Speed | Training Mem | Inference Mem |
|---|---|---|---|---|---|
| GTM | 35.73M | 0.290s/iter | 0.165s/iter | 8324.00MB | 1250.00MB |
| Time-Moe(base) | 50.00M | 0.840s/iter | 0.095s/iter | 1812.48MB | 226.70MB |
| GPT-2(6)-768 | 82.28M | 0.104s/iter | 0.054s/iter | 5230.00MB | 2566.00MB |
| FEDformer-768 | 30.75M | 0.467s/iter | 0.172s/iter | 9535.00MB | 1880.19MB |
| TimesNet-768 | 42.21M | 1.849s/iter | 0.547s/iter | 35871.00MB | 1904.18MB |

Table 10: Analysis of model inference latency and computational overhead in critical modules. **F.A.** denotes Fourier Attention module.

| GPU | Channel | Inference (s/item) | FFT+iFFT (s/item) | F.A. (s/item) |
|---|---|---|---|---|
| A100 | 1 | 0.043 | 0.0007 | 0.033 |
| | 7 | 0.044 | 0.0007 | 0.034 |
| | 862 | 0.142 | 0.0009 | 0.103 |
| RTX4090 | 1 | 0.041 | 0.0007 | 0.031 |
| | 7 | 0.041 | 0.0007 | 0.031 |
| | 862 | 0.144 | 0.0009 | 0.107 |

Table 11: Analysis of model inference latency across different frequency modules.

| Low-rank modules | Channel | Inference (s/item) | F.A. (s/item) |
|---|---|---|---|
| 1 | 1 | 0.030 | 0.020 |
| | 7 | 0.030 | 0.020 |
| 10 | 1 | 0.060 | 0.049 |
| | 7 | 0.061 | 0.050 |
| 20 | 1 | 0.092 | 0.080 |
| | 7 | 0.094 | 0.081 |

and efficiency. As shown in Table 9, GTM achieves suitable trade-offs for industrial deployment: it ranks second in parameter size(35.7M), training speed(0.290s/iter for batchsize 128), and inference memory(1.25GB), and retains competitive performance in inference speed(0.165s/iter) and training memory(8.32GB), demonstrating both efficiency and applicability for real-time deployment.

We further break down the computational overhead of the Fourier Attention module and FFT/iFFT operations. Table 10 presents latency measured on both A100 and RTX4090 GPUs. For univariate data (1440 input points, 96 prediction length), GTM achieves a total inference latency of just 0.043s/item, with FFT/iFFT and Fourier Attention modules introducing only marginal overhead. Similar results are observed for the multivariate case(ETT and Traffic data), confirming GTM's low-latency and capable for sub-second real-time streaming applications. We provide more model efficiency scale analysis with significantly longer prediction lengths in Appendix B.3.8.

Finally, we evaluate how the number of low-rank modules in Fourier Attention affects inference latency. Table 11 shows that increasing the number of modules provides finer-grained distribution representation across temporal granularities, with only a gradual, sub-linear rise in processing time. Even with 20 modules, latency stays below 0.1s/item, meeting the demands of real-time applications. This highlights GTM's ability to balance model expressiveness and computational efficiency.

## 5 CONCLUSION

Large-scale TS analysis poses distinct challenges compared to LLMs, particularly in learning effective universal knowledge and building models for multi-task settings. In this paper, we propose GTM, a general framework for TS analysis that utilizes a decoder-only architecture. GTM incorporates granularity-aware attention mechanisms in both the temporal and frequency domains to improve TS representations. Furthermore, we introduce a blank infilling pre-training strategy specifically designed for multi-task TS analysis, unifying all generative downstream tasks. Experimental results show that GTM either matches or outperforms SOTA methods across all generative TS analysis tasks. Additionally, our findings demonstrate that GTM adheres to scaling laws, achieving better performance with larger model sizes and more extensive pre-training datasets. However, challenges and limitations remain in the design of TSFM, including the lack of large-scale datasets and the absence of consistent benchmark models and settings. A detailed discussion of future work and limitations is provided in Appendix B.5 for further enhancement.

## 6 ACKNOWLEDGEMENTS

The work was supported by grants from the National Natural Science Foundation of China (Nos. 92367110 and U23A20319).

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

## A    THE USE OF LARGE LANGUAGE MODELS

LLMs were used only occasionally to help polish the writing (propose new words, grammar and spelling correction). All technical ideas, experimental designs, analyses, conclusions, writing were developed and carried out entirely by the authors. The authors have full responsibility for the final text.

## B    TECHNICAL APPENDICES AND SUPPLEMENTARY MATERIAL

### B.1    ADDITIONAL RELATED WORK

**LLM Empowered TSFMs**: This line of works follow the paradigm that freeze LLM encoder backbones while simultaneously fine-tuning/adapting the input and projection heads for forecasting, and notable ones include Time-LLM(Jin et al., 2024), LLM4TS(Chang et al., 2024), GTP4TS(Zhou et al., 2023), UniTime(Liu et al., 2024a), Chronos(Ansari et al., 2024) and Tempo(Cao et al., 2024). This effectiveness of this paradigm is currently under debate in the sense that some works present promising results while the latest ablation studies show the counterpart (Tan et al., 2024).

Table 12 highlights the key distinctions between our approach and existing SOTA models. **First**, whereas prior TSFMs primarily rely on temporal information from discrete scalar values, our method uniquely integrates both temporal and frequency-domain features through a Fourier attention mechanism that captures time granularity-aware representations. **Second**, previous models often require downstream task-specific customization at the token, pre-training strategy, or model level. In contrast, our approach introduces a hybrid masking-based pre-training strategy that unifies reconstruction and autoregressive objectives, enabling generative-task-agnostic adaptation without additional modifications.

### B.2    DETAILS OF IMPLEMENTATION AND EXPERIMENTAL SETTINGS

#### B.2.1    DATASETS DESCRIPTION

We use the UTSD-12G dataset, released by (Liu et al., 2024b), for pre-training. The Unified Time Series Dataset (UTSD) includes seven domains: Energy, Environment, Health, IoT, Nature, Transportation, and Web, with varying sampling frequencies. It contains up to 1 billion time points and hierarchical structures, supporting large-scale time series model research. The overall statistics of UTSD-12G is shown in Table 13.

For downstream tasks such as long-term forecasting and imputation, we conduct experiments on five widely used public datasets from (Wu et al., 2021): ETTh, ETTm, Weather, Electricity, and Traffic. For anomaly detection, we utilize five popular datasets: SMD (Su et al., 2019), MSL, SMAP (Hundman et al., 2018), SWaT (Mathur & Tippenhauer, 2016), and PSM (Abdulaal et al., 2021). For classification, we select ten standard datasets from Bagnall et al. (2018): EthanolConcentration, FaceDetection, Handwriting, Heartbeat, JapaneseVowels, PEMS-SF, SelfRegulationSCP1, SelfRegulationSCP2, SpokenArabicDigits, and UWaveGestureLibrary. Dataset statistics for these tasks are summarized in Tables 14, 15, and16. Among these datasets, the ETTm dataset represents the longest-range testing scenario, spanning over 725 days and containing up to 69,680 time points at a 15-minute sampling interval.

#### B.2.2    BASELINE MODEL SELECTION

We summarize the baseline models in Table17. We classify these models into four categories, including LLM-enhanced models for TS analysis, MLP-based models, Transformer-based models, and TSFMs. The TSFMs are further divided into two sub-categories: task-specific foundation models and multi-task foundation models. Since each model has its own design goals and experimental settings, it is challenging to align them all for reproducing their best results presented in papers. Therefore, we follow established protocols from previous works and select typical models as benchmarks for each downstream task, ensuring a fair comparison of GTM with SOTA results.

Table 12: Comparison between GTM and SOTA time series foundation models trained from scratch. The models are characterized by their approach to representation learning, ability to handle downstream tasks, and adaptability to multi-task scenarios. The list of the abbreviation of the table is: Temporal Domain: **T. D.**, Frequency Domain: **F. D.**, Anomaly Detection: **AD.**, Inference Adaption: **Inf. Ad.**

| | Time Series Features | | | Downstream Tasks | | | | Adaptability |
|---|---|---|---|---|---|---|---|---|
| | **T. D.** | **F. D.** | Time Gran. | Forecasting | **AD.** | Imputation | CLF. | W/o Inf. Ad. |
| PatchTST, Lag-Llama, GPD GPHT, TimesFM, MOIRAI, UTSD, TTMs, TIME-MOE | ✓ | ✗ | ✗ | ✓ | ✗ | ✗ | ✗ | ✗ |
| TimeSiam, LPTM | ✓ | ✗ | ✗ | ✓ | ✗ | ✗ | ✓ | ✗ |
| TIMER, UP2ME | ✓ | ✗ | ✗ | ✓ | ✓ | ✓ | ✗ | ✗ |
| UniTS | ✓ | ✗ | ✗ | ✓ | ✓ | ✓ | ✓ | ✗ |
| GTM(ours) | ✓ | ✓ | ✓ | ✓ | ✓ | ✓ | ✓ | ✓ |

Table 13: Statistics of UTSD-12G dataset.

| Domain | Dataset Number | Time Points | File Size | Freq. |
|---|---|---|---|---|
| Energy | 3 | 175.06M | 4334M | [4 sec, 30 min, Hourly] |
| Environment | 3 | 31.54M | 286M | [Hourly] |
| Health | 9 | 289.72M | 2685M | [1ms, 2ms, 4ms, 8ms] |
| IoT | 1 | 165.4M | 2067M | [20ms] |
| Nature | 11 | 241.4M | 2227M | [33ms, Hourly, 3h, Daily] |
| Transport | 1 | 3.13M | 72M | [Hourly] |
| Web | 1 | 116.49M | 388M | [Daily] |

Table 14: Statistics of datasets for forecasting & imputation.

| Dataset | Length | Dimension | Frequency |
|---|---|---|---|
| ETTh | 17420 | 7 | 1 hour |
| ETTm | 69680 | 7 | 15 min |
| Weather | 52696 | 21 | 10 min |
| Electricity | 26304 | 321 | 1 hour |
| Traffic | 17544 | 862 | 1 hour |

### B.2.3 EXPERIMENTAL SETTINGS AND IMPLEMENTATION DETAILS

**Pre-training** In the pre-training stage, we trained our GTM model on the UTSD-12G dataset (Liu et al., 2024b). During data preprocessing, we defined a lookback window of 1440 timestamps and split the raw data into overlapping samples with a stride $\tau = 192$. We then generated 15 patches with a patch size $L_p = 96$. To enable the model to learn both reconstruction and forecasting objectives, for each training instance, we empirically set the hyperparameter $pred\_ratio$ to 0.3, and masked the last 30% of the sequence (tail masking) with probability $pred\_ratio$. For other critical model hyperparameters, we set the batch size to 1024 and the learning rate to $1 \times 10^{-5}$, using Adam as the optimizer with a cosine annealing learning rate decay. We trained for 30 epochs with an early stopping mechanism, and the decay steps were proportional to the number of training epochs. In the model backbone, we set the number of layers (N-stack) to 12 and the feature dimension to 768. The Fourier Knowledge Attention layer consisted of 5 attention modules, each with a low-rank matrix parameterized by $AB$, where $A \in \mathbb{R}^{385 \times 1}$, $B \in \mathbb{R}^{1 \times 385}$. We provide pseudo-code of GTM architecture and pre-training strategy in Algorithm 1. Finally, we implemented the GTM model in PyTorch (Paszke et al., 2019) and trained it on 6 NVIDIA A100 40GB GPUs.

**Fine-tune** We present experimental settings for three generative downstream tasks.

- **Long-term Forecasting** For long-term forecasting, we directly reuse the pre-trained GTM model without any special adaptations, only removing the masking process. We dynamically choose look-back window in range $[96, 1440]$ and forecast future time points $T \in \{96, 192, 336, 720\}$. The results are compared with the best-performing results SOTA models presented in papers or source codes.

Table 15: Statistics of datasets for anomaly detection.

| Dataset | Training size | Validation size | Test size | Dimension | Frequency | Anomaly rate |
|---------|---------------|-----------------|-----------|-----------|-----------|--------------|
| MSL | 46653 | 11664 | 73729 | 55 | 1 min | 10.5% |
| SMAP | 108146 | 27037 | 427617 | 25 | 1 min | 12.8% |
| SMD | 566724 | 141681 | 708420 | 38 | 1 min | 4.2% |
| SWaT | 396000 | 99000 | 449919 | 51 | 1 sec | 12.1% |
| PSM | 105984 | 26497 | 87841 | 25 | 1 min | 27.8% |

Table 16: Statistics of datasets for classification.

| Dataset | Train Cases | Test Cases | Dimensions | Length | Classes |
|---------|-------------|------------|------------|--------|---------|
| EthanolConcentration | 261 | 263 | 3 | 1751 | 4 |
| FaceDetection | 5890 | 3524 | 144 | 62 | 2 |
| Handwriting | 150 | 850 | 3 | 152 | 26 |
| Heartbeat | 204 | 205 | 61 | 405 | 2 |
| JapaneseVowels | 270 | 370 | 12 | 29 | 9 |
| PEMS-SF | 267 | 173 | 963 | 144 | 7 |
| SelfRegulationSCP1 | 268 | 293 | 6 | 896 | 2 |
| SelfRegulationSCP2 | 200 | 180 | 7 | 1152 | 2 |
| SpokenArabicDigits | 6599 | 2199 | 13 | 93 | 10 |
| UWaveGestureLibrary | 120 | 320 | 3 | 315 | 8 |

Table 17: Selected SOTA baseline models for downstream tasks comparison.

| Task | Method Types | Method |
|------|--------------|--------|
| Forecasting | LLM-Enhanced for TS | GPT4TS |
| | MLP-based | DLinear |
| | Transformer-based | PatchTST, FEDformer, Autoformer, Informer |
| | task-specific foundation model | TTMs UTSD |
| | multi-task foundation model | UniTS-SUP, UniTS-PMT, TimesNet |
| Anomaly Detection | LLM-Enhanced for TS | GPT4TS |
| | MLP-based | DLinear |
| | Transformer-based | PatchTST, FEDformer, Autoformer, Informer |
| | task-specific foundation model | / |
| | multi-task foundation model | TimesNet, UP2ME |
| Imputation | LLM-Enhanced for TS | GPT4TS |
| | MLP-based | DLinear |
| | Transformer-based | PatchTST, FEDformer, Autoformer Informer |
| | task-specific foundation model | UTSD |
| | multi-task foundation model | TimesNet UP2ME |
| Classification | LLM-Enhanced for TS | GPT4TS |
| | MLP-based | / |
| | Transformer-based | iTransformer |
| | task-specific foundation model | / |
| | multi-task foundation model | UniTS-SUP, UniTS-PMT, TimesNet |

- **Imputation** To align with benchmark settings, we follow the protocol proposed by (Zhou et al., 2023) for imputation tasks. We use point-wise missing ratios of

---

**Algorithm 1** GTM pre-train strategy

---

**Require:** : Input look_back time series $x$, $x \in \mathbb{R}^{L \times C}$; look_back window length $L$, number of channels or variables $C$; number of patches $N_p$; patch length $L_p$; no. of masked patch $N_{mp}$; no. of reconstruction patch label $N_{rp}$; no. of total patches $N_{total} = N_{mp} + N_{rp}$; patch embedding dimension D.

1: ▷ **CI** and **Patching**:
2: $P = \textbf{Patch}(\textbf{CI}(x))$             ▷ $P \in \mathbb{R}^{N_p \times L_p}$
3: ▷ **Masking**:
4: $X_{in} = [P_{crpt}, S_{in}]$             ▷ $X_{in} \in \mathbb{R}^{(N_{mp}+N_{rp}) \times L_p}$
5: ▷ **Embedding**:
6: $H_{in}^0 = Embedding(X_{in})$           ▷ $H_{in}^0 \in \mathbb{R}^{N_{total} \times D}$
7: **for** $i = 1$ **to** $N$ **do**            ▷ through GTM blocks
8:     ▷ Temporal attention:
9:     $H_{TemAttOut}^{n-1} = SelfAttention(H_{in}^{n-1})$     ▷ $H_{TemAttOut}^{n-1} \in \mathbb{R}^{N_{total} \times D}$
10:    ▷ Fourier attention:
11:    $H_{out}^n = FourierAttention(H_{TemAttOut}^{n-1})$     ▷ $H_{out}^n \in \mathbb{R}^{N_{total} \times D}$
12: **end for**
13: ▷ Output Projection:
14: $X_{out} = MLP(H_{out}^N)$           ▷ $X_{out} \in \mathbb{R}^{N_{total} \times L_p}$
15: **return** $X_{out}$

---

$\{12.5\%, 25\%, 37.5\%, 50\%\}$ at the time-point level for interpolation, omitting the patching process. For all other aspects, we reuse the settings from the pre-training stage.

- **Anomaly Detection** We use a common adjustment strategy (Xu et al., 2018; Su et al., 2019; Shen et al., 2020) for anomaly detection: if an anomaly is detected at any time point in an abnormal segment, all anomalies in that segment are considered detected. This approach is based on the fact that detecting one abnormal point usually triggers an alert for the entire segment in real-world scenarios. We calculate F1-scores for each datasets to evaluate the results. the As we do in other generative tasks, we directly reuse the GTM model settings from the pre-training stage.

- **Classification** For discriminative tasks such as classification, we replace the projection head to output class label probabilities instead of future time step predictions, while keeping the rest of the model architecture unchanged to ensure smoothly adaptation. We employ cross-entropy loss, aiming to minimize the divergence between the predicted and true class distributions, which is equivalent to maximizing the log-likelihood of the correct label. Model performance is evaluated using accuracy, enabling direct comparison between GTM and SOTA TSFMs.

### B.3 FULL RESULTS

Due to space limitations in the main body of the paper, we provide the full experimental results in this section, to complement the discussion in section 4.

### B.3.1 FORECASTING

Table 18 demonstrates the full results of long-term forecasting. it shows that GTM outperforms all the SOTA models, achieving the best result in 21 and second best in 22 out of total 50 tests. The second best model PatchTST, achieves the best in 14 and second best in 15.

Table 18: Full results of MSE and MAE for long-term forecasting. We conduct experiments for different length $T \in \{96, 192, 336, 720\}$. **Bold** and underline numbers denote the best and the 2nd-best results, respectively.

| Models | | GTM | | GPT4TS | | UniTS-PMT | | TTM$_E$ | | PatchTST | | TimesNet | | DLinear | | FEDformer | | Autoformer | | Informer | |
|---|---|---|---|---|---|---|---|---|---|---|---|---|---|---|---|---|---|---|---|---|---|
| Dataset | $T$ | MSE | MAE | MSE | MAE | MSE | MAE | MSE | MAE | MSE | MAE | MSE | MAE | MSE | MAE | MSE | MAE | MSE | MAE | MSE | MAE |
| ETTh1 | 96 | **.360** | **.398** | .376 | **.397** | .390 | .411 | .363 | - | .370 | .400 | .384 | .402 | .375 | .399 | .376 | .415 | .435 | .446 | .865 | .713 |
| | 192 | .397 | .422 | .416 | .418 | .432 | .438 | **.394** | - | .413 | .429 | .436 | .429 | .405 | **.416** | .423 | .446 | .456 | .457 | 1.008 | .792 |
| | 336 | .420 | .437 | .442 | **.433** | .480 | .460 | **.403** | - | .422 | .440 | .491 | .469 | .439 | .443 | .444 | .462 | .486 | .487 | 1.107 | .809 |
| | 720 | **.438** | .457 | .477 | **.456** | .542 | .508 | .449 | - | .447 | .468 | .521 | .500 | .472 | .490 | .469 | .492 | .515 | .517 | 1.181 | .865 |
| | Avg | .404 | .429 | .427 | **.426** | .461 | .454 | **.402** | - | .413 | .434 | .458 | .450 | .422 | .437 | .428 | .453 | .473 | .476 | 1.040 | .795 |
| ETTm1 | 96 | **.282** | **.341** | .292 | .346 | - | - | .293 | - | .293 | .346 | .338 | .375 | .299 | .343 | .326 | .390 | .510 | .492 | .672 | .571 |
| | 192 | **.325** | .366 | .332 | .372 | - | - | .335 | - | .333 | .370 | .374 | .387 | .335 | **.365** | .365 | .415 | .514 | .495 | .795 | .669 |
| | 336 | **.353** | **.385** | .366 | .394 | - | - | .364 | - | .369 | .392 | .410 | .411 | .369 | .386 | .392 | .425 | .510 | .492 | 1.212 | .871 |
| | 720 | **.396** | **.410** | .417 | .421 | - | - | .408 | - | .416 | .420 | .478 | .450 | .425 | .421 | .446 | .458 | .527 | .493 | 1.166 | .823 |
| | Avg | **.339** | **.376** | .352 | .383 | - | - | .350 | - | .352 | .382 | .400 | .406 | .357 | .378 | .382 | .422 | .515 | .493 | .961 | .734 |
| Weather | 96 | **.147** | **.197** | .162 | .212 | .157 | .206 | .154 | - | .149 | .198 | .172 | .220 | .176 | .237 | .238 | .314 | .249 | .329 | .300 | .384 |
| | 192 | **.192** | **.241** | .204 | .248 | .208 | .251 | .207 | - | .194 | .241 | .219 | .261 | .220 | .282 | .325 | .370 | .325 | .370 | .598 | .544 |
| | 336 | .250 | .291 | .254 | .286 | .264 | .291 | .250 | - | **.245** | **.282** | .280 | .306 | .265 | .319 | .351 | .391 | .351 | .391 | .578 | .523 |
| | 720 | **.310** | **.334** | .326 | .337 | .344 | .344 | .324 | - | .314 | .334 | .365 | .359 | .323 | .362 | .415 | .426 | .415 | .426 | 1.059 | .741 |
| | Avg | **.225** | .266 | .237 | .270 | .243 | .273 | .234 | - | .225 | **.263** | .259 | .287 | .246 | .300 | .332 | .375 | .335 | .379 | .634 | .548 |
| Traffic | 96 | **.351** | .250 | .388 | .282 | .465 | .298 | .372 | - | .360 | **.249** | .593 | .321 | .410 | .282 | .576 | .359 | .597 | .371 | .719 | .391 |
| | 192 | .373 | .260 | .407 | .290 | .484 | .306 | **.365** | - | .379 | **.256** | .617 | .336 | .423 | .287 | .610 | .380 | .607 | .382 | .696 | .379 |
| | 336 | .388 | .267 | .412 | .294 | .494 | .312 | **.379** | - | .392 | **.264** | .629 | .336 | .436 | .296 | .608 | .375 | .623 | .387 | .777 | .420 |
| | 720 | .428 | .288 | .450 | .312 | .534 | .335 | **.425** | - | .432 | **.286** | .640 | .350 | .466 | .315 | .621 | .375 | .639 | .395 | .864 | .472 |
| | Avg | **.385** | .266 | .414 | .294 | .494 | .313 | .385 | - | .390 | **.263** | .620 | .336 | .433 | .295 | .603 | .372 | .616 | .383 | .764 | .416 |
| Electricity | 96 | .131 | .225 | .139 | .238 | .157 | .258 | **.129** | - | .129 | **.222** | .168 | .272 | .140 | .237 | .186 | .302 | .196 | .313 | .274 | .368 |
| | 192 | .149 | .243 | .153 | .251 | .173 | .272 | .148 | - | **.147** | **.240** | .184 | .289 | .153 | .249 | .197 | .311 | .211 | .324 | .296 | .386 |
| | 336 | .166 | **.259** | .169 | .266 | .185 | .284 | .161 | - | **.163** | **.259** | .198 | .300 | .169 | .267 | .213 | .328 | .214 | .327 | .300 | .394 |
| | 720 | .201 | .292 | .206 | .297 | .219 | .314 | **.193** | - | .197 | **.290** | .220 | .320 | .203 | .301 | .233 | .344 | .236 | .342 | .373 | .439 |
| | Avg | .161 | .254 | .167 | .263 | .184 | .282 | **.158** | - | .159 | **.252** | .192 | .295 | .166 | .263 | .207 | .321 | .214 | .326 | .311 | .397 |

We further conduct error bar analysis by running 10 independent trials for long-term forecasting tasks. The 95% confidence interval for each metric is calculated as $t_{0.025,\,n-1} \times \frac{\text{std}}{\sqrt{n}}$, where $t_{0.025,\,n-1}$ is the 97.5th percentile of the Student's $t$-distribution with $n-1$ degrees of freedom. For $n = 10$ runs, it is approximately 2.26. As shown in Table 19, the error bars for both MSE and MAE across all prediction lengths and datasets are consistently low, indicating high reliability and stability of the reported results.

To provide a comprehensive baseline comparison for long-term forecasting tasks, we further include two recent TSFMs, Sundial-SmallLiu et al. (2025) and Time-MOE-BaseShi et al. (2024), both of which are comparable to GTM in model size and are evaluated under similar experimental settings. Note that Time-MOE-Base was not evaluated on the Electricity dataset; therefore, best count statistics are reported both for all tasks and for the subset where Time-MOE-Base results are available. As shown in Table 20, GTM achieves the best performance on 13 out of 20 metrics overall, and on 8 out of 15 metrics when directly compared with Time-MOE-Base (numbers in parentheses in the table), slightly outperforming the Time-MOE-Base model. In contrast, Sundial-Small achieves the best result in only one case. These results demonstrate GTM's strong competitiveness and robustness across diverse datasets and prediction horizons.

For more experiments on challenging, real-world datasets, we have identified two such kind of datasets that have not yet been over exploited: one is an open PV(PhotoVoltaic) solar energy forecasting datasetCarreira Pedro et al. (2019), and the other is the L2C (lead-to-cash) datasetSaha et al. (2024), which combines observations of Business Key Performance Indicators (Biz-KPIs) and IT events. We also have reproduced two SOTA models, PatchTST and TimesNet, conducting experiments on forecasting with various prediction length. Table 21 shows that GTM consistently delivers SOTA performance across both the PV and L2C datasets, achieving the best results in most test cases for both MSE and MAE metrics. The most significant improvement of GTM over competing methods is observed on the L2C dataset with a prediction length of 720. For MSE, GTM achieves a score of 0.7170, outperforming the second-best method (TimesNet, 1.2984) which means a 44.8% reduction.

Table 19: 95% CI error bar analysis for forecasting tasks.

| Dataset | pred_len | MSE | MSE error-bar | MAE | MAE error-bar |
|---------|----------|-----|---------------|-----|---------------|
| ETTh1 | 96 | 0.3611 | ±0.00093 | 0.3991 | ±0.00072 |
| | 192 | 0.3990 | ±0.00099 | 0.4241 | ±0.00099 |
| | 336 | 0.4236 | ±0.00099 | 0.4395 | ±0.00099 |
| | 720 | 0.4428 | ±0.00344 | 0.4643 | ±0.00265 |
| | Avg. | 0.4066 | ±0.00157 | 0.4318 | ±0.00136 |
| ETTm1 | 96 | 0.2828 | ±0.0014 | 0.3430 | ±0.0010 |
| | 192 | 0.3304 | ±0.0022 | 0.3698 | ±0.0005 |
| | 336 | 0.3580 | ±0.0021 | 0.3890 | ±0.0009 |
| | 720 | 0.4040 | ±0.0020 | 0.4122 | ±0.0016 |
| | Avg. | 0.3438 | ±0.0019 | 0.3785 | ±0.0010 |
| Weather | 96 | 0.1474 | ±0.00043 | 0.1983 | ±0.00050 |
| | 192 | 0.1943 | ±0.00107 | 0.2427 | ±0.00115 |
| | 336 | 0.2445 | ±0.00014 | 0.2876 | ±0.00021 |
| | 720 | 0.3100 | ±0.00229 | 0.3355 | ±0.00115 |
| | Avg. | 0.2241 | ±0.00099 | 0.2660 | ±0.00079 |

For MAE, GTM attains a value of 0.5218 compared to PatchTST's 0.8460, resulting in a 38.3% reduction.

### B.3.2 IMPUTATION

Table22 provides the full results of Imputation for various data missing ratios of $\{12.5\%, 25\%, 37.5\%, 50\%\}$ at the time-point level. Except for the Electricity dataset (where it achieved second-best performance), GTM outperforms all other methods in other experiments.

### B.3.3 EXTENDED ANOMALY DETECTION

Recent work by (Liu & Paparrizos, 2024) has highlighted several critical challenges in time series anomaly detection, including flawed datasets and biased evaluation metrics. To provide a more comprehensive evaluation of our model, we utilize the TSB-AD benchmark, which features an extensive and carefully curated collection of datasets, widely used measures, along with broad coverage of TSFMs testing results. Table 23 presents the mean accuracy scores across the TSB-AD-U datasets using various evaluation metrics. Compared to SOTA TSFMs such as MOMENT, TimesFM, Lag-Llama, Chronos, and other deep learning models, GTM achieves the best performance in 8 out of 9 metrics. These results demonstrate that GTM is highly adaptable and robust across diverse datasets and evaluation criteria.

### B.3.4 EFFECTIVENESS OF PRE-TRAINING

**Forecasting** Table 24 presents a detailed comparison between the pre-trained GTM model and the GTM model without pre-training. We also conduct experiments for different length $T \in \{96, 192, 336, 720\}$. The results demonstrate that pre-trained GTM model outperforms the non-pre-trained version, highlighting the benefit of the pre-training stage in leveraging general knowledge from large-scale datasets.

**Imputation** Table 25 provides detailed results of comparison in Imputation tasks between the pre-trained GTM model and the GTM model without pre-training. As described in Sec4.3, we also conduct experiment for different data missing ratios of $\{12.5\%, 25\%, 37.5\%, 50\%\}$ at the time-point

Table 20: Full MSE and MAE results for long-term forecasting on additional SOTA TSFMs: Time-MOE-Base and Sundial-Small. Experiments are conducted for prediction lengths $T \in \{96, 192, 336, 720\}$. **Bold** numbers denote the best results.

| Models | | | GTM | | Time-MOE-b | | Sundial-s | |
|---|---|---|---|---|---|---|---|---|
| dataset | Pred_len | MSE | MAE | MSE | MAE | MSE | MAE |
| ETTh1 | 96 | 0.360 | 0.398 | 0.345 | **0.373** | **0.341** | 0.381 |
| | 192 | 0.397 | 0.422 | **0.372** | **0.396** | 0.381 | 0.408 |
| | 336 | 0.420 | 0.437 | **0.389** | **0.412** | 0.405 | 0.424 |
| | 720 | 0.438 | 0.457 | **0.410** | **0.443** | 0.433 | 0.458 |
| | Avg. | 0.404 | 0.429 | **0.379** | **0.406** | 0.390 | 0.418 |
| ETTm1 | 96 | **0.282** | 0.341 | 0.286 | **0.334** | 0.292 | 0.342 |
| | 192 | 0.325 | 0.366 | **0.307** | **0.358** | 0.337 | 0.376 |
| | 336 | **0.353** | **0.385** | 0.354 | 0.390 | 0.370 | 0.401 |
| | 720 | **0.396** | **0.410** | 0.433 | 0.445 | 0.418 | 0.433 |
| | Avg. | **0.339** | **0.376** | 0.345 | 0.381 | 0.354 | 0.388 |
| weather | 96 | **0.147** | **0.197** | 0.151 | 0.203 | 0.158 | 0.206 |
| | 192 | **0.192** | **0.241** | 0.195 | 0.246 | 0.205 | 0.253 |
| | 336 | 0.250 | 0.291 | **0.247** | **0.288** | 0.254 | 0.290 |
| | 720 | **0.310** | **0.334** | 0.352 | 0.366 | 0.315 | 0.336 |
| | Avg. | **0.225** | **0.266** | 0.236 | 0.275 | 0.233 | 0.271 |
| Electricity | 96 | **0.131** | **0.225** | - | - | 0.134 | 0.231 |
| | 192 | **0.149** | **0.243** | - | - | 0.154 | 0.251 |
| | 336 | **0.166** | **0.259** | - | - | 0.174 | 0.271 |
| | 720 | **0.201** | **0.292** | - | - | 0.215 | 0.307 |
| | Avg. | **0.161** | **0.254** | - | - | 0.169 | 0.265 |
| Best Count | | **13(8)** | **12(7)** | 6 | 8 | 1 | 0 |

level. As expected, the pre-trained GTM model outperforms the non-pre-trained version in all tests, achieving significant improvements.

### B.3.5 ABLATION TEST

Table 26 presents the full ablation results for forecasting tasks with varying prediction lengths, includes $T \in \{96, 192, 336, 720\}$ time points. The comparison involves the complete GTM model, an advanced version of GTM without the frequency knowledge attention module, and a baseline version that includes only the temporal analysis module. The results demonstrate that the complete design of the GTM model effectively supports the learning of universal representations for MTS datasets with varying time granularities.

### B.3.6 SCALABILITY ANALYSIS

Figure 6 illustrates the data scalability analysis, while Table 27 provides the complete forecasting results using different pre-trained data sizes. The findings confirm that GTM follows scaling laws, where pre-training on larger datasets consistently enhances fine-tuning performance across diverse downstream tasks.

Table 21: Full results of MSE and MAE for long-term forecasting on PhotoVoltaic(PV) and Lead-to-cash(L2C) datasets. **Bold** numbers denote the best results.

| Dataset | Pred_len | GTM | | PatchTST | | TimesNet | |
|---|---|---|---|---|---|---|---|
| | | MSE | MAE | MSE | MAE | MSE | MAE |
| L2C | 96 | 0.4463 | 0.3508 | 0.3516 | **0.3143** | **0.3330** | 0.3575 |
| | 240 | 0.7692 | **0.5359** | 0.7732 | 0.5670 | **0.7345** | 0.5598 |
| | 720 | **0.7170** | **0.5218** | 1.3400 | 0.8460 | 1.2984 | 0.8374 |
| | Avg. | **0.6442** | **0.4695** | 0.8216 | 0.5758 | 0.7886 | 0.5849 |
| PV | 60 | **0.1763** | **0.2504** | 0.2017 | 0.2892 | 0.2578 | 0.3921 |
| | 240 | **0.3030** | **0.3880** | 0.3650 | 0.4691 | 0.3655 | 0.4691 |
| | 720 | 0.5476 | 0.5616 | 0.5780 | 0.5852 | **0.4928** | **0.5414** |
| | Avg. | **0.3423** | **0.4000** | 0.3816 | 0.4478 | 0.3720 | 0.4675 |
| | Best_count | **5** | **6** | 0 | 1 | 3 | 1 |

Table 22: Full results of Imputation. We conduct experiment for different data missing ratios of $\{12.5\%, 25\%, 37.5\%, 50\%\}$ at the time-point level.

| Models | | GTM | | GPT4TS | | TimesNet | | PatchTST | | DLinear | | Fedformer | | Informer | |
|---|---|---|---|---|---|---|---|---|---|---|---|---|---|---|---|
| Dataset | Mask Ratio | MSE | MAE | MSE | MAE | MSE | MAE | MSE | MAE | MSE | MAE | MSE | MAE | MSE | MAE |
| ETTh1 | 12.5% | **.034** | **.125** | .043 | .140 | .057 | .159 | .093 | .201 | .151 | .267 | .070 | .190 | .114 | .234 |
| | 25% | **.046** | **.143** | .054 | .156 | .069 | .178 | .107 | .217 | .180 | .292 | .106 | .236 | .140 | .262 |
| | 37.5% | **.059** | **.163** | .072 | .180 | .084 | .196 | .120 | .230 | .215 | .318 | .124 | .258 | .174 | .293 |
| | 50% | **.073** | **.179** | .107 | .216 | .102 | .215 | .141 | .248 | .257 | .347 | .165 | .299 | .215 | .325 |
| | Avg. | **.053** | **.152** | .069 | .173 | .078 | .187 | .115 | .224 | .201 | .306 | .117 | .246 | .161 | .279 |
| ETTm1 | 12.5% | **.015** | **.082** | .017 | .085 | .023 | .101 | .041 | .130 | .080 | .193 | .052 | .166 | .063 | .180 |
| | 25% | **.019** | **.090** | .022 | .096 | .023 | .101 | .044 | .135 | .080 | .193 | .052 | .166 | .063 | .180 |
| | 37.5% | **.023** | **.100** | .029 | .111 | .029 | .111 | .049 | .143 | .103 | .219 | .069 | .191 | .079 | .200 |
| | 50% | **.029** | **.112** | .040 | .128 | .036 | .124 | .055 | .151 | .132 | .248 | .089 | .218 | .093 | .218 |
| | Avg. | **.021** | **.096** | .028 | .105 | .027 | .107 | .047 | .140 | .093 | .206 | .062 | .177 | .071 | .188 |
| Weather | 12.5% | .026 | .046 | .026 | .049 | **.025** | **.045** | .029 | .049 | .039 | .084 | .041 | .107 | .218 | .326 |
| | 25% | .030 | .055 | **.028** | **.052** | .029 | .052 | .031 | .053 | .048 | .103 | .064 | .163 | .219 | .326 |
| | 37.5% | **.031** | **.057** | .033 | .060 | .031 | .057 | .035 | .058 | .057 | .117 | .107 | .229 | .222 | .328 |
| | 50% | **.034** | **.061** | .037 | .065 | .034 | .062 | .038 | .063 | .066 | .134 | .183 | .312 | .228 | .331 |
| | Avg. | **.030** | **.054** | .031 | .056 | .030 | .054 | .060 | .144 | .052 | .110 | .099 | .203 | .222 | .328 |
| Electricity | 12.5% | .077 | .191 | .080 | .194 | .085 | .202 | **.055** | **.160** | .092 | .214 | .107 | .237 | .037 | .093 |
| | 25% | .084 | .199 | .087 | .203 | .089 | .206 | **.065** | **.175** | .118 | .247 | .120 | .251 | .042 | .100 |
| | 37.5% | .090 | .206 | .094 | .211 | .094 | .213 | **.076** | **.189** | .144 | .276 | .136 | .266 | .049 | .111 |
| | 50% | .096 | .215 | .101 | .220 | .100 | .221 | **.091** | **.208** | .175 | .305 | .158 | .284 | .053 | .114 |
| | Avg. | .086 | .202 | .090 | .207 | .092 | .210 | **.072** | **.183** | .132 | .260 | .130 | .259 | .045 | .104 |

### B.3.7 HYPER-PARAMETER ANALYSIS

The look-back window length and patch length are two critical hyperparameters in the GTM model. We conducted experiments with varying values for these parameters to analyze the model's sensitivity. Table 28 shows that performance steadily improves as the patch length increases, while Table 29 demonstrates that both MAE and MSE results are consistently enhanced as the look-back window length is extended.

Table 23: Summary accuracy comparison of mean value on TSB-AD-U by various metrics. The best-performing method as per each metric is marked in **bold**.

| Models\Metrics | AUC-PR | AUC-ROC | VUS-PR | VUS-ROC | Standard-F1 | PA-F1 | Event-based-F1 | R-based-F1 | Affiliation-F1 |
|---|---|---|---|---|---|---|---|---|---|
| GTM | **0.33** | **0.71** | 0.36 | **0.78** | **0.38** | **0.86** | **0.71** | **0.36** | **0.91** |
| MOMENT (FT) | 0.30 | 0.69 | **0.39** | 0.76 | 0.35 | 0.65 | 0.49 | 0.35 | 0.86 |
| TimesFM | 0.28 | 0.67 | 0.3 | 0.74 | 0.34 | 0.84 | 0.63 | 0.34 | 0.89 |
| Lag-Llama | 0.25 | 0.65 | 0.27 | 0.72 | 0.3 | 0.77 | 0.59 | 0.31 | 0.88 |
| Chronos | 0.26 | 0.66 | 0.27 | 0.73 | 0.32 | 0.83 | 0.61 | 0.33 | 0.88 |
| TimesNet | 0.18 | 0.61 | 0.26 | 0.72 | 0.24 | 0.67 | 0.47 | 0.21 | 0.86 |
| FITS | 0.17 | 0.61 | 0.26 | 0.73 | 0.23 | 0.65 | 0.42 | 0.2 | 0.86 |
| AnomalyTransformer | 0.08 | 0.5 | 0.12 | 0.56 | 0.12 | 0.53 | 0.34 | 0.14 | 0.77 |

Table 24: Full results of forecasting comparison between GTM and GTM w/o pre-train. We conduct experiments for different length $T \in \{96, 192, 336, 720\}$.

| Models | | | GTM | | GTM w/o pretrain | |
|---|---|---|---|---|---|---|
| Dataset | pred_len | | MSE | MAE | MSE | MAE |
| ETTh1 | 96 | | **0.360** | **0.398** | 0.376 | 0.412 |
| | 192 | | **0.397** | **0.422** | 0.411 | 0.428 |
| | 336 | | **0.420** | **0.437** | 0.454 | 0.453 |
| | 720 | | **0.438** | **0.457** | 0.500 | 0.497 |
| | Avg. | | **0.404(+7.1%)** | **0.429(+4.0%)** | 0.435 | 0.447 |
| ETTm1 | 96 | | **0.282** | **0.341** | 0.291 | 0.352 |
| | 192 | | **0.325** | **0.366** | 0.335 | 0.378 |
| | 336 | | **0.353** | **0.385** | 0.366 | 0.397 |
| | 720 | | **0.396** | **0.410** | 0.415 | 0.429 |
| | Avg. | | **0.339(+3.3%)** | **0.376(3.3%)** | 0.351 | 0.389 |
| Weather | 96 | | **0.147** | **0.197** | 0.154 | 0.204 |
| | 192 | | **0.192** | **0.241** | 0.212 | 0.267 |
| | 336 | | **0.250** | **0.291** | 0.275 | 0.323 |
| | 720 | | **0.310** | **0.334** | 0.337 | 0.365 |
| | Avg. | | **0.225(+7.8%)** | **0.266(+8.0%)** | 0.244 | 0.289 |
| Traffic | 96 | | **0.351** | **0.250** | 0.353 | 0.252 |
| | 192 | | **0.373** | 0.260 | 0.373 | **0.259** |
| | 336 | | **0.388** | **0.267** | 0.391 | 0.270 |
| | 720 | | **0.428** | **0.288** | 0.432 | 0.291 |
| | Avg. | | **0.385(+0.5%)** | **0.266(+0.8%)** | 0.387 | 0.268 |
| Electricity | 96 | | **0.131** | **0.225** | 0.132 | 0.225 |
| | 192 | | **0.149** | **0.243** | 0.150 | 0.244 |
| | 336 | | **0.166** | **0.259** | 0.170 | 0.262 |
| | 720 | | **0.201** | **0.292** | 0.203 | 0.294 |
| | Avg. | | **0.161(+1.2%)** | **0.254(+0.8%)** | 0.163 | 0.256 |

### B.3.8 MODEL EFFICIENCY ANALYSIS

We further analyze model efficiency by conducting experiments with longer prediction lengths and larger input look-back windows. As shown in Table 30, the inference time remains nearly constant even as both the look-back window and prediction length increase by an order of magnitude. This illustrates that GTM does not fully saturate the computational resources of the A100 GPU, demonstrating high efficiency at the current scales and is well-suited for practical deployment in real-world sub-second streaming applications.

Table 25: Full results of Imputation comparison between GTM and GTM w/o pre-training. We conduct experiments for varying data missing ratios of $\{12.5\%, 25\%, 37.5\%, 50\%\}$ at the time-point level.

| Models | | | GTM | | GTM w/o pretrain | |
|---|---|---|---|---|---|---|
| Dataset | Mask Ratio | MSE | MAE | MSE | MAE |
| ETTh1 | 12.5% | **0.034** | **0.125** | 0.037 | 0.131 |
| | 25% | **0.046** | **0.143** | 0.048 | 0.146 |
| | 37.5% | **0.059** | **0.163** | 0.060 | 0.163 |
| | 50% | **0.073** | **0.179** | 0.077 | 0.184 |
| | Avg. | **0.053(+3.6%)** | **0.152(+2.5%)** | 0.055 | 0.156 |
| ETTm1 | 12.5% | **0.015** | **0.082** | 0.020 | 0.096 |
| | 25% | **0.019** | **0.090** | 0.019 | 0.091 |
| | 37.5% | **0.023** | **0.100** | 0.024 | 0.101 |
| | 50% | **0.029** | **0.112** | 0.030 | 0.113 |
| | Avg. | **0.021(+8.6%)** | **0.096(+4.0%)** | 0.023 | 0.100 |
| Weather | 12.5% | **0.026** | **0.046** | 0.028 | 0.051 |
| | 25% | **0.030** | **0.055** | 0.029 | 0.056 |
| | 37.5% | **0.031** | **0.057** | 0.032 | 0.060 |
| | 50% | **0.034** | **0.061** | 0.049 | 0.088 |
| | Avg. | **0.030(+11.7%)** | **0.054(+14.2%)** | 0.034 | 0.063 |
| Electricity | 12.5% | **0.077** | **0.191** | 0.078 | 0.192 |
| | 25% | **0.084** | **0.199** | 0.084 | 0.199 |
| | 37.5% | **0.090** | **0.206** | 0.091 | 0.207 |
| | 50% | **0.096** | **0.215** | 0.097 | 0.215 |
| | Avg. | **0.086(+1.2%)** | **0.202(+0.5%)** | 0.087 | 0.203 |

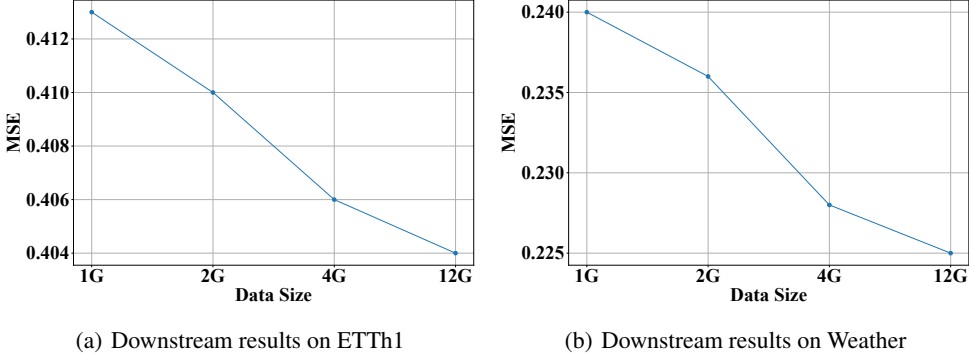

(a) Downstream results on ETTh1      (b) Downstream results on Weather

Figure 6: Data scalability analysis. GTM achieves better results with larger pre-training datasets.

From an architectural perspective, there are three mainstream output projection designs in time series forecasting models. Below we clarify these designs and discuss their implications for flexibility and inference efficiency:

- **Flatten layer with a linear projection (direct mapping)**

  In this design, the backbone outputs a tensor of size $[B, N_p, D]$ (batch size $B$, number of patches $N_p$, feature dimension $D$), which is flattened and projected via a linear layer of shape $[N_p \times D, L]$, where $L$ is the prediction length. This approach is adopted by models such as PatchTST, TimesNet, Crossformer, FreTS, etc..

  *Limitations*: The output head must be reconfigured for each $L$, limiting flexibility for variable-length forecasting. It is a clear disadvantage for TSFMs. Moreover, inference time increases with larger $L$ due to the growing size of the output head.

- **Autoregressive Approach**

Table 26: Full results of ablation test in forecasting tasks. Experiments are conducted for varying prediction lengths, includes $T \in \{96, 192, 336, 720\}$ time points.

| Models | | GTM | | GTM w/o time_gran. | | GTM w/o Freq. | |
|---|---|---|---|---|---|---|---|
| Dataset | Pred_len | MSE | MAE | MSE | MAE | MSE | MAE |
| ETTh1 | 96 | **0.360** | **0.398** | 0.372 | 0.406 | 0.384 | 0.416 |
| | 192 | **0.397** | **0.422** | 0.405 | 0.427 | 0.408 | 0.429 |
| | 336 | **0.420** | **0.437** | 0.428 | 0.437 | 0.433 | 0.443 |
| | 720 | **0.438** | **0.457** | 0.450 | 0.463 | 0.449 | 0.466 |
| | Avg. | **0.404(3.57%, 2.42%)** | **0.429(2.28%,0.92%)** | 0.414(1.19%) | 0.433(1.37%) | 0.419 | 0.439 |
| ETTm1 | 96 | **0.282** | **0.341** | 0.299 | 0.353 | 0.301 | 0.354 |
| | 192 | **0.325** | **0.366** | 0.334 | 0.372 | 0.335 | 0.375 |
| | 336 | **0.353** | **0.385** | 0.360 | 0.391 | 0.363 | 0.393 |
| | 720 | **0.396** | **0.410** | 0.398 | 0.411 | 0.398 | 0.412 |
| | Avg. | **0.339(2.87%,2.59%)** | **0.376(2.08%,1.57%)** | 0.348(0.29%) | 0.382(0.52%) | 0.349 | 0.384 |
| Weather | 96 | **0.147** | **0.197** | 0.153 | 0.217 | 0.158 | 0.212 |
| | 192 | **0.192** | **0.241** | 0.206 | 0.254 | 0.208 | 0.258 |
| | 336 | **0.250** | **0.291** | 0.252 | 0.293 | 0.256 | 0.297 |
| | 720 | **0.310** | **0.334** | 0.311 | 0.335 | 0.313 | 0.337 |
| | Avg. | **0.225(3.43%,2.60%)** | **0.266(3.62%,3.27%)** | 0.231(0.86%) | 0.275(0.36%) | 0.233 | 0.276 |
| Traffic | 96 | **0.351** | **0.250** | 0.355 | 0.253 | 0.359 | 0.256 |
| | 192 | **0.373** | **0.260** | 0.374 | 0.262 | 0.379 | 0.264 |
| | 336 | **0.388** | **0.267** | 0.389 | 0.270 | 0.393 | 0.271 |
| | 720 | **0.428** | **0.288** | 0.431 | 0.291 | 0.435 | 0.293 |
| | Avg. | **0.385(1.79%,0.52%)** | **0.266(1.85%,1.12%)** | 0.387(1.28%) | 0.269(0.74%) | 0.392 | 0.271 |
| Electricity | 96 | **0.131** | **0.225** | 0.132 | 0.226 | 0.134 | 0.227 |
| | 192 | **0.149** | **0.243** | 0.150 | 0.246 | 0.152 | 0.248 |
| | 336 | **0.166** | **0.259** | 0.168 | 0.262 | 0.169 | 0.264 |
| | 720 | **0.201** | **0.292** | 0.202 | 0.295 | 0.205 | 0.296 |
| | Avg. | **0.161(2.42%,1.23%)** | **0.254(1.93%,1.17%)** | 0.163(1.21%) | 0.257(0.77%) | 0.165 | 0.259 |

Table 27: Full results of scalability analysis on pre-trained data size in forecasting tasks. Experiments are conducted for varying prediction lengths, includes $T \in \{96, 192, 336, 720\}$ time points.

| Data_size | | 1G | | 2G | | 4G | | 12G | |
|---|---|---|---|---|---|---|---|---|---|
| | Pred_len | MSE | MAE | MSE | MAE | MSE | MAE | MSE | MAE |
| ETTh1 | 96 | 0.372 | 0.409 | 0.369 | 0.405 | 0.363 | 0.400 | **0.360** | **0.398** |
| | 192 | 0.404 | 0.425 | 0.405 | 0.426 | 0.399 | 0.423 | **0.397** | **0.422** |
| | 336 | 0.427 | 0.439 | 0.423 | 0.438 | 0.422 | 0.438 | **0.420** | **0.437** |
| | 720 | 0.448 | 0.462 | 0.445 | 0.459 | 0.441 | 0.458 | **0.438** | **0.457** |
| | Avg. | 0.413 | 0.434 | 0.410 | 0.432 | 0.406 | 0.429 | **0.404** | **0.429** |
| Weather | 96 | 0.147 | 0.197 | 0.148 | 0.199 | 0.147 | 0.198 | **0.147** | **0.197** |
| | 192 | 0.193 | 0.244 | 0.192 | 0.241 | 0.193 | 0.242 | **0.192** | **0.241** |
| | 336 | 0.257 | 0.295 | 0.253 | 0.292 | 0.251 | 0.291 | **0.250** | **0.291** |
| | 720 | 0.364 | 0.361 | 0.351 | 0.352 | 0.321 | 0.340 | **0.310** | **0.334** |
| | Avg. | 0.240 | 0.274 | 0.236 | 0.271 | 0.228 | 0.267 | **0.225** | **0.266** |

In this approach, the model predicts one future value at a time: at each step $t$, it uses its previous prediction $\hat{y}_{t-1}$ together with the input history to predict $\hat{y}_t$. This process is repeated until the desired prediction length $L$ is reached.

*Advantage*: Enables high flexibility, the same output head can generate forecasts of varying lengths without retraining.

Table 28: Performance of GTM for Different Patch Lengths.

| Patch-len | 8 | | 16 | | 32 | | 64 | | 96 | |
|---|---|---|---|---|---|---|---|---|---|---|
| Dataset | MSE | MAE | MSE | MAE | MSE | MAE | MSE | MAE | MSE | MAE |
| ETTh1 | 0.426 | 0.443 | 0.416 | 0.441 | 0.422 | 0.439 | 0.413 | 0.437 | **0.405** | **0.429** |
| ETTm1 | 0.363 | 0.402 | 0.349 | 0.381 | 0.355 | 0.388 | 0.351 | 0.379 | **0.342** | **0.377** |

Table 29: Performance of GTM for different look-back window lengths.

| Seq-len | 96 | | 192 | | 336 | | 512 | | 672 | | 1440 | |
|---|---|---|---|---|---|---|---|---|---|---|---|---|
| Dataset | MSE | MAE | MSE | MAE | MSE | MAE | MSE | MAE | MSE | MAE | MSE | MAE |
| ETTh1 | 0.435 | 0.452 | 0.428 | 0.447 | 0.416 | 0.439 | 0.418 | 0.440 | 0.411 | 0.433 | **0.405** | **0.429** |
| ETTm1 | 0.371 | 0.401 | 0.363 | 0.395 | 0.355 | 0.389 | 0.354 | 0.387 | 0.342 | 0.379 | **0.342** | **0.377** |

Table 30: Model efficiency analysis for varying prediction and look-back window lengths.

| GPU | Channels | Lookback Len. | Pred. Len. | Inference (s/item) | FFT + iFFT (s/item) | Fourier Attention (s/item) |
|---|---|---|---|---|---|---|
| A100 | 1 | 1440 | 96 | 0.043 | 0.0007 | 0.033 |
| | 1 | 2880 | 1440 | 0.043 | 0.0007 | 0.033 |
| | 1 | 5120 | 2880 | 0.043 | 0.0007 | 0.033 |
| | 1 | 14400 | 5120 | 0.043 | 0.0007 | 0.033 |

*Limitations*: Inference latency scales linearly with $L$ (since prediction is done step by step), and error may accumulate as the prediction length increases. For these reasons, SOTA TSFMs rarely use this mechanism for output projection.

- **Sequence to Sequence(seq2seq) approach**

  Here, the model's projection layer is designed to directly output the entire prediction sequence of arbitrary length. In our implementation, the backbone output $[B, N_p, D]$ is processed to generate $N_{\text{pred}} = N_p \times \text{patchsize}$ time points, corresponding to the look-back window. At post-processing, outputs are truncated to the required prediction length $L$.

  *Advantages*: Offers flexible output lengths, since the output head does not require specific configuration for each $L$, making it highly suitable for variable-length forecasting. Inference time is generally insensitive to $L$, as the whole sequence is produced in parallel. This design explains why, in our tests (with a fixed look-back window), inference latency remains nearly constant for different prediction lengths up to the input window length. SOTA TSFMs such as TIMER, UP2ME, UniTS etc., adopt this approach.

  *Note*: the distinction between the seq2seq and autoregressive approaches can sometimes be ambiguous: for example, while TIMER follows a seq2seq implementation, its paper describes the output generation process as "autoregressive".

## B.4 VISUALIZATION ANALYSIS

### B.4.1 DISTRIBUTION DISCREPANCY OF TS DATASETS

We conduct measurement analysis on UTSD-12G datasets and 5 popular multi-domain datasets for downstream tasks as described in Table 13 and 14. To complement the limited information available in the temporal domain, we transform the datasets into the frequency domain using FFT. This allows us to analyze data distribution patterns from various perspectives, including amplitude, phase, periodicity, frequency resolution, etc.. Due to the complexity of the joint distribution, we apply a non-parametric estimation method, specifically 2-D Kernel Density Estimation (KDE) (Equation 12), to estimate the joint probability density distribution (PDF) of amplitude-frequency and phase-frequency for time series data with varying granularities. We use a 2-D Gaussian kernel

function (Equation 13) and 2-D Scott's rule (Equation 14) as bandwidth fuction. Where $n$ denotes number of data samples, $h$ is the bandwidth, $\sigma$ and $\mu$ are standard deviation and mean of the samples. The results are presented in Figure 1. It reveals notable discrepancies in the joint distributions across TS datasets with different time granularities. This observation highlights the importance of learning these distribution discrepancies as critical knowledge in the process of building a universal representation of MTS, which has often been overlooked in previous studies.

$$\hat{f}(x, y) = \frac{1}{n h_x h_y} \sum_{i=1}^{n} K\left(\frac{x - x_i}{h_x}, \frac{y - y_i}{h_y}\right) \tag{12}$$

$$K(x, y) = \frac{1}{2\pi\sigma_x\sigma_y} \exp\left(-\frac{(x - \mu_x)^2}{2\sigma_x^2} - \frac{(y - \mu_y)^2}{2\sigma_y^2}\right) \tag{13}$$

$$h_x = h_y = n^{-\frac{1}{6}}(\sigma_x\sigma_y)^{\frac{1}{2}} \tag{14}$$

### B.4.2 LONG-TERM FORECASTING

To clearly present the results, we select some representative samples for visualization analysis. Figure7 shows the long-term forecasting results from 4 different datasets. We select 3 typical forecasting results from 3 different dimensions of each datasets.

### B.4.3 IMPUTATION

Figure 8 illustrates the imputation results from three dimensions across four different datasets. Clearly, GTM can effectively reconstruct the missing data, adapting to varying data patterns.

### B.4.4 ANOMALY DETECTION

Figure 9 demonstrates four anomaly events detected by GTM in two datasets, along with their corresponding anomaly scores. The results align precisely with the labeled anomalies in the data.

### B.5 LIMITATIONS AND FUTURE WORK

Although GTM achieves promising results in multi-task time series analysis, several important limitations remain. First, the current architecture is primarily effective for data exhibiting clear periodicity or trend, while its robustness to low signal-to-noise ratio (SNR) or highly irregular time series is not yet fully understood. Future work will focus on developing a frequency-domain time granularity-aware learning module and expanding GTM into a comprehensive Mixture-of-Experts (MoE) framework with gate control mechanisms, aiming to further enhance its representation learning capacity and adaptability to complex temporal patterns. In addition, we plan to leverage the GIFT (Aksu et al., 2024), a larger-scale time series dataset for pre-training and utilize GIFT-Eval for downstream task evaluation, which will provide a more rigorous and diverse assessment of GTM's generalization ability. However, the absence of unified evaluation protocols and benchmarks—where algorithms are compared under consistent pre-training datasets, hyperparameter settings and experimental conditions—remains a significant barrier to fair and reproducible research in the field. Addressing these challenges, including improving model robustness and establishing standardized benchmarking practices, will be crucial for advancing time series analysis and realizing the full potential of GTM in both academic and real-world scenarios.

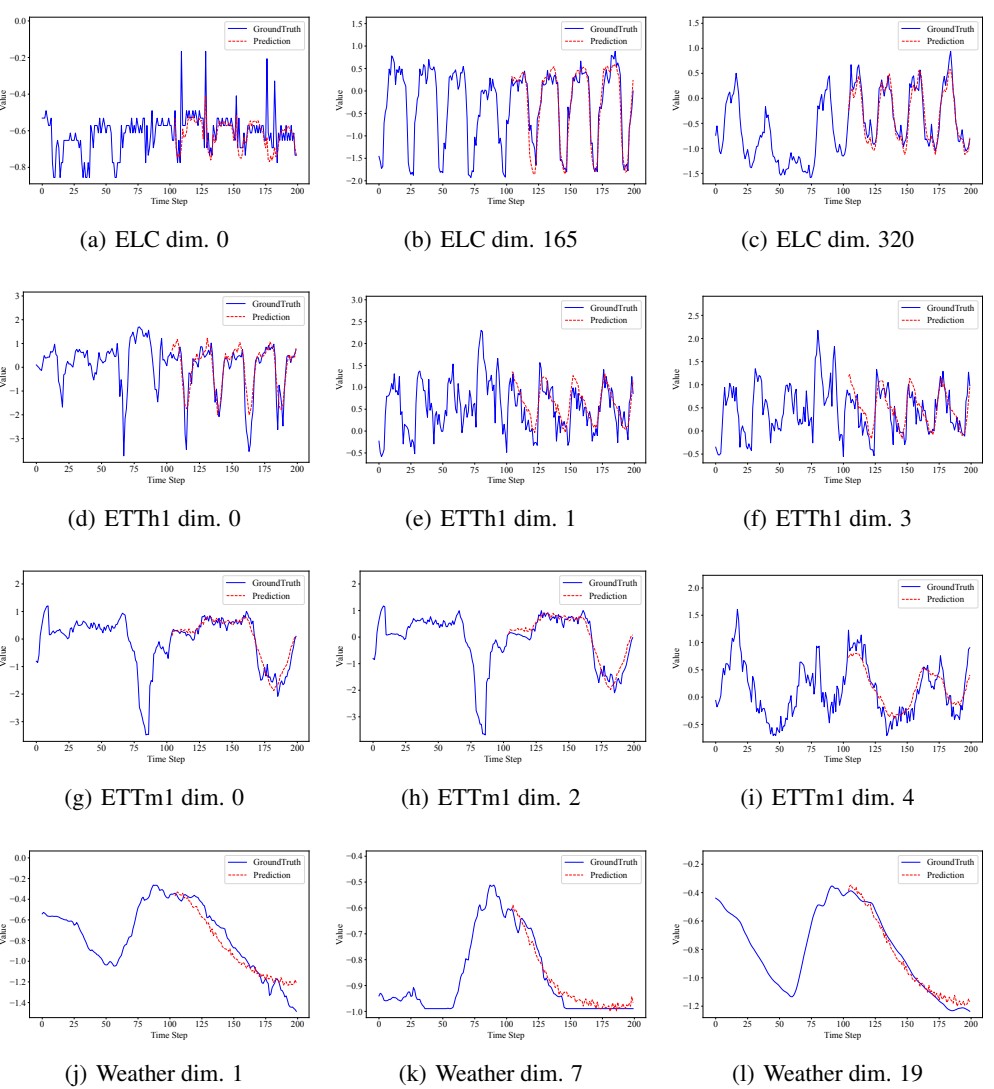

Figure 7: Visualization of forecasting results.

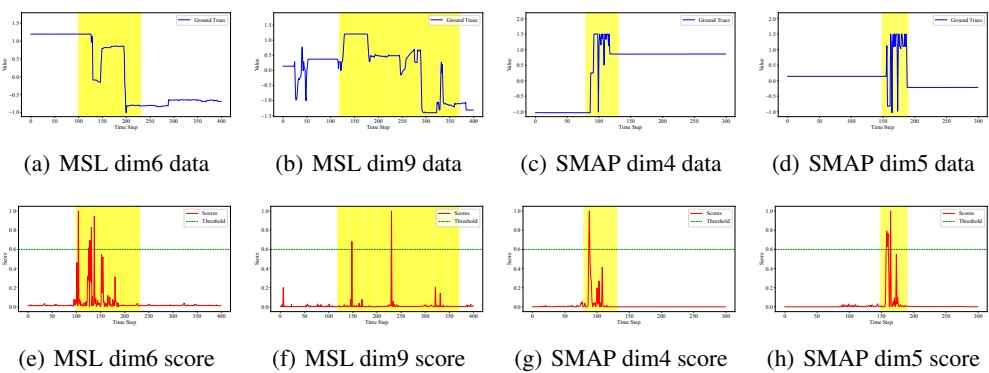

Figure 9: Visualization of anomaly_detection results.

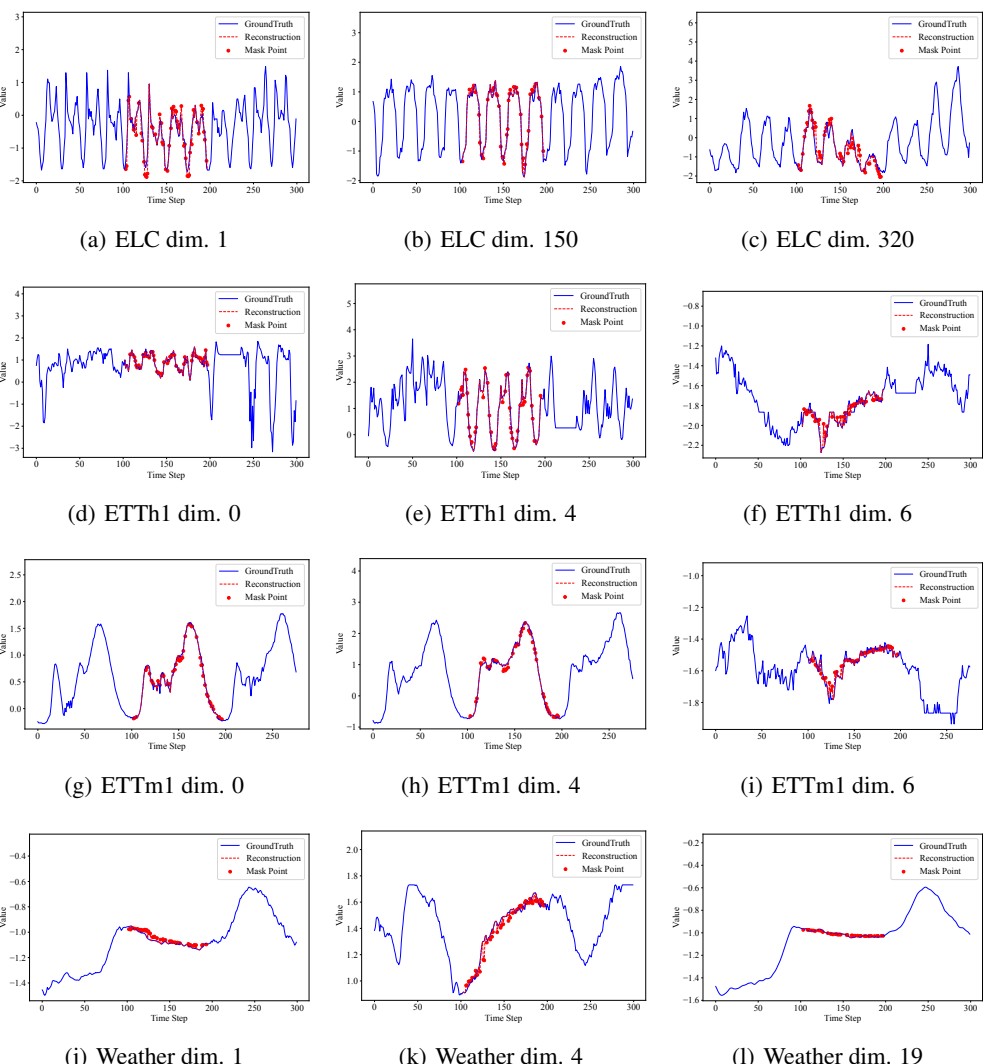

Figure 8: Visualization of imputation results.

