# OpenReview forum: "GTM: A General Time-series Model for Enhanced Representation Learning of Time-Series data"
_ICLR.cc/2026/Conference — ICLR 2026 Poster_

### Official Review · Reviewer_boeL · 2025-10-28

**Soundness:** 2
**Presentation:** 3
**Contribution:** 3
**Rating:** 6
**Confidence:** 4

**Summary:**

The paper proposes a time-series model that incorporates frequency-domain information in self-attention, a pre-training strategy that combines reconstruction and autoregressive objectives, and it shows that this model (after being trained on large dataset) it can be considered as a foundation model. Experimental results on a large number of time series datasets for different tasks shows that the model is competitive to existing ones, and in many cases leads to higher performance.

The main argument for the improved performance of the proposed model is the use of frequency domain features, something that the existing models are not exploiting. This argument has been used in many cases in the past, and it does not seem to be well-justified: no theoretical explanation is provided to prove the advantage of using a specific spate-time domain transformation (the Fourier transform) for internal neural network representations, while the experimental comparisons do not show a clear boost in performance that would be attributed to the adoption of a clearly better type of information.

Overall, the experimental analysis is comprehensive and it shows advances for the proposed model.

**Strengths:**

- The proposed model shows to achieve comparable or even better (on average) performance compared to existing models in a variety of tasks and datasets.
 - A comprehensive experimental evaluation is provided.

**Weaknesses:**

- No computational and memory cost analyses are provided for the proposed model.
 - No comparisons in terms of inference time are provided for the proposed model in relation to the competition.
 - Comparisons with recent related models, e.g., UniTS, Moirai, ST-LLM, are not provided.

**Questions:**

- What is the computational and memory costs of the proposed model?
 - How does the model compare with existing ones in inference speed?
 - Comparisons with more recent related models.

---

> ### Author Response · Authors · 2025-11-20
>
> Thank you for your valuable feedback regarding the model efficiency analysis and SOTA model comparisons. We address each of your comments in detail below:
> 1. Computational and memory cost analysis
> - We have conducted efficiency analyses for GTM, comparing it with four baseline models (Time-MOE, GPT-2(6)-768, TimesNet-768, FEDformer-768), as detailed in Table 24, Appendix B.3.8. GTM ranks second in parameter size(35M), training speed(0.290s/iter) and inference memory(1.25GB), and third in inference speed (0.165s/iteration for batch size 128) and training memory(8.32GB), demonstrating GTM's suitability for real-time industrial deployment.
> - Additionally, we analyzed the computational overhead of Fourier Attention and FFT/iFFT operations. As shown in Table 1, on both A100 and RTX 4090 GPUs, GTM achieves fast inference: for univariate time series, total latency is only 0.043s per item, with FFT/iFFT and Fourier Attention contributing minimal overhead. Similar results are observed for multivariate data, further confirming GTM’s low-latency, real-time capability.
> - Table 1: Analysis of model inference latency and computational overhead in critical modules
> |GPU|Channels|Inference time(s/item)|FFT+iFFT(s/item)|Fourier Attention (s/item)|
> |-|-|-|-|-|
> |A100-40GB|1|0.043|0.0007|0.033|
> ||7|0.044|0.0007|0.034|
> ||862|0.142|0.0009|0.103|
> |RTX 4090 24GB|1|0.041|0.0007|0.031|
> ||7|0.041|0.0007|0.031|
> ||862|0.144|0.0009|0.107|
>
> 2. Comparisons with recent related models, e.g., UniTS, Moirai, ST-LLM
>
> We have presented our results in comparison with SOTA models such as Moirai and UniTS in the paper.
>
> Regarding Moirai, it is important to note that this model primarily emphasizes zero-shot performance, and its zero-shot results are often compared with the full-shot results of other SOTA models. To ensure a relatively fair comparison, we referenced results reproduced by Timer (ICML 2024) and selected the Moirai-s model, as both our model and Moirai-s are pre-trained on the same UTSD open datasets contributed by Timer and have similar model sizes. Table 5 in Section 4.6 reports the average zero-shot forecasting results across various datasets. It shows that GTM outperforms Moirai on 4 out of 5 datasets, and achieves the best overall zero-shot performance compared to other SOTA TSFMs.
> - Table 5 Zero-shot capability (MSE) of GTM compared to SOTA TSFMs
> | Dataset | GTM| TIMER-1B| MOIRAI-S | MOMENT | TimesFM| CHRONOS-S1 |
> |-|-|-|-|-|-|-|
> | ETTh1| **0.407**| 0.438 | 0.441 | 0.674| 0.414 | 0.571|
> | ETTm1| 0.593 | 0.690 | 0.562 | 0.670| **0.354** | 0.632|
> | weather | **0.172** | 0.181 | 0.195 | 0.255| -| - |
> | ECL| **0.187** | 0.192 | 0.212 | 0.744| -| - |
> | Traffic | 0.542 | **0.458** | 0.616 | 1.293| -| - |
> | Average | **0.380** | 0.392 | 0.405 | 0.727| -| - |
>
> Regarding UniTS, GTM was compared against it on both long-term forecasting and classification tasks under similar experimental settings. Detailed results are provided in Table 1 (Section 4.2) and Table 4 (Section 4.5) of the paper; we list the relevant results here for further comparison.
> - Forecasting: As shown in Table 1, GTM consistently outperforms UniTS-PMT across all test datasets in terms of average MSE, demonstrating superior forecasting accuracy.
> - Table 1: Avg. MSE & MAE forecasting results
>
> | Models | GTM   |   | UniTS-PMT |   |
> | -| - | - | - | - |
> | dataset| MSE   | MAE   | MSE   | MAE   |
> | ETTh1  | **.404**  | **.429**  | .461  | .454  |
> | weather| **.225**  | **.266**  | .243  | .273  |
> | traffic| **.385**  | **.266**| .494  | .313  |
> | Electricity| **.161**  | **.254**| .184  | .282|
>
> - Classification: Table 4 summarizes the accuracy results for seven classification tasks. GTM achieves the best performance in six out of seven tasks, often with a substantial margin, while UniTS-PMT leads in only one task with a slight advantage.
>
> - Table 4  The Accuracy results of Classification tasks
> |Dataset/Model| GTM| UNITS-SUP | UNITS-PMT | GPT4TS| TimesNet| iTransformer |
> |-|-|-|-|-|-|-|
> |EthanolConcentration | 34.2| / | / | 34.2| **35.7**| 28.1 |
> |FaceDetection| **69.9** | 65.4| 58| 69.2| 68.6| 66.3 |
> |Handwriting| **34.8** | / | / | 32.7| 32.1| 24.2 |
> |Heartbeat| 77.5| 63.9| 65.4| 77.2| **78**| 75.6 |
> |Japanese Vowels| 92.1| 92.2| 90.3| **98.6** | 98.4| 96.6 |
> | PEMS-SF| 88.4| 83.2| 82.7| 87.9| **89.6** | 87.9 |
> | SelfRegulationSCP1| 92.5| / | / | **93.2** | 91.8| 90.2 |
> | SelfRegulationSCP2| **60**| 48.9| 57.2| 59.4| 57.2| 54.4 |
> | SpokenArabicDigits|**99.2** | 96.8| 95.5| 99.2| 99| 96|
> | UWaveGestureLibrary|**89.3** | 82.2| 85.3| 88.1| 85.3| 85.9 |
> | Best Count| **5**|0|0|2|3|0|
>
> Regarding ST-LLM, we identified two models from the literature survey. But it seems that both are not directly related to time series analysis. Please let us know if we have referenced the wrong works. Thank you very much.
> - [1] Ruyang Liu, et al. "ST-LLM: Large Language Models Are Effective Temporal Learners"
> - [2] Chenxi Liu, et al. "Spatial-Temporal Large Language Model for Traffic Prediction"

---

### Official Review · Reviewer_mmJe · 2025-10-29

**Soundness:** 3
**Presentation:** 3
**Contribution:** 3
**Rating:** 8
**Confidence:** 3

**Summary:**

The paper introduces the General Time-series Model , a novel foundation model based on the empirical finding that time-series data exhibits distinct frequency-domain distributions  at different temporal granularities . To exploit this, GTM incorporates a novel Fourier attention mechanism that learns to capture these granularity-aware representations. A key contribution is its unified pre-training strategy, which uses a hybrid masking approach to jointly optimize reconstruction and autoregressive objectives. This makes GTM the first generative-task-agnostic model, enabling it to handle diverse tasks like forecasting, imputation, and anomaly detection without any task-specific modifications. Extensive experiments confirm that GTM consistently outperforms or matches state-of-the-art models on these generative tasks, achieves strong results on classification, and demonstrates effective scaling with increased model size and data.

**Strengths:**

1. The paper is exceptionally well-motivated. It begins with a clear, empirical analysis using FFT and 2D KDE to demonstrate that the joint probability distributions of amplitude-frequency and phase-frequency vary significantly across different temporal granularities . This finding provides a strong, intuitive justification for the core architectural novelty of the proposed model.

2. The paper introduces a novel Fourier attention mechanism designed to explicitly capture the granularity-aware features identified in the initial analysis. Instead of just applying a standard FFT, the module learns to weigh different frequency-domain transformations based on the input's time granularity. This is a clever method to inject domain-specific knowledge about time-scale properties directly into the learning process.

3. A key contribution is the novel pre-training strategy that unifies reconstruction and autoregressive objectives. By adopting a GLM-style hybrid masking approach—which combines random span masking  with consecutive tail masking —the model is trained to handle a variety of generative tasks from the outset.

4. This unified pre-training strategy successfully establishes GTM as a generative-task-agnostic model. The authors demonstrate that the same pre-trained model can be applied to forecasting, imputation, and anomaly detection without any task-specific architectural modifications, which is a significant step toward a true "foundation model" for time series.

**Weaknesses:**

1. The hybrid masking strategy is a core contribution, but a critical detail is underspecified. The paper mentions applying a "controlled proportion of consecutive [MASK] tokens to at the tail". The value or control mechanism for this proportion, which presumably balances the reconstruction and autoregressive objectives, is not defined in the main paper or the appendix's implementation details .

2. The proposed Fourier attention module adds a significant number of operations within each decoder block: a standard temporal attention, an FFT, a granularity-based attention, five low-rank matrix multiplications, one full-matrix multiplication, and an inverse FFT . While the authors commendably provide an efficiency analysis in Table 24, which shows GTM is competitive, this added complexity is a non-trivial tradeoff for the performance gains.

**Questions:**

Listed in Weakness

---

> ### Author Response · Authors · 2025-11-20
>
> Thank you for your constructive feedback. Please find our detailed responses below.
> 1. Lack of specification for hybrid masking proportion and control mechanism
>
> Thank you for pointing out the lack of a detailed description of our hybrid masking strategy and its control mechanism. We provide our implementation details as follows:
> - In our implementation, we introduce a hyperparameter, pred_ratio, to control the proportion of consecutive [MASK] tokens applied to the tail of the sequence. Specifically, for each training instance, we generate a random number uniformly in [0, 1]. If this number is less than or equal to pred_ratio, we apply a consecutive mask to the tail (e.g., masking the last 30% of the sequence). Otherwise, we use a random masking strategy.
> - The pred_ratio hyperparameter provides a flexible and interpretable way to balance between the reconstruction (random masking) and autoregressive forecasting (consecutive tail masking) objectives. We select its value empirically based on validation performance before the pre-training stage, and in our experiments, we found that setting pred_ratio between 0.2 and 0.3 yields stable and robust results across datasets. This aligns with our experimental settings in GTM paper, where the proportion of autoregressive masking is typically set in the same range to ensure sufficient exposure to both objectives during pre-training. Importantly, this hybrid masking mechanism, enables GTM to be seamlessly adaptable to a wide range of downstream generative tasks without any modification, and to outperform SOTA models on all these tasks. We will add these details in the revised pdf.
>
> 2. Non-trivial Computational Overhead Introduced by Fourier Attention Module
> - To address concerns regarding computational overhead, we conducted supplementary experiments to measure the latency introduced by the Fourier Attention module and FFT/iFFT operations. These analyses were performed on both Nvidia A100 and RTX 4090 GPUs. As shown in Table 1, for univariate time series, the total inference latency is only 0.043 seconds per item, with FFT/iFFT and Fourier Attention modules contributing just 0.0007 seconds and 0.033 seconds per item, respectively. Comparable low-latency results were observed for multivariate cases, indicating the model’s cost-effectiveness and suitability for real-time industrial deployment.
> - Table 1: Analysis of model inference latency and computational overhead in critical modules
> |GPU|Channels|Inference time(s/item)|FFT+iFFT(s/item)|Fourier Attention (s/item)|
> |-|-|-|-|-|
> |A100-40GB|1|0.043|0.0007|0.033|
> ||7|0.044|0.0007|0.034|
> ||862|0.142|0.0009|0.103|
> |RTX 4090 24GB|1|0.041|0.0007|0.031|
> ||7|0.041|0.0007|0.031|
> ||862|0.144|0.0009|0.107|
>
> - We further evaluated the impact of increasing the number of low-rank modules in the Fourier Attention mechanism on inference latency (see Table 2, under the same experimental settings). The results show that adding more modules results in a gradual, sub-linear increase in processing time. Notably, even with 20 modules, latency remains below 0.1 seconds per item, meeting real-time applications' requirements. This demonstrates that GTM can flexibly balance model expressiveness and computational speed. For future work, we plan to extend Fourier Attention to a large-scale MoE architecture and explore distributed, parallel processing to further reduce latency.
> - Table 2 Trade-off analysis between low-rank modules and inference latency
> |Low-rank modules |Channels |Inference time(s/item) |Fourier Attention(s/item) |
> |-|-|-|-|
> |1|1|0.030|0.020|
> ||7| 0.030| 0.020|
> |5|1| 0.043| 0.033|
> ||7| 0.044| 0.034|
> |10 |1| 0.060| 0.049|
> ||7 | 0.061| 0.050|
> |15 |1| 0.076| 0.065|
> ||7| 0.078| 0.066|
> |20|1| 0.092|0.080|
> ||7| 0.094| 0.081|

---

### Official Review · Reviewer_bvHo · 2025-11-01

**Soundness:** 2
**Presentation:** 4
**Contribution:** 2
**Rating:** 4
**Confidence:** 3

**Summary:**

This work introduces GTM, a general model for generative time series learning. The authors claim that this model is one of the first to be task-agnostic for generative tasks, allowing the model to be applied in a variety of settings without task-specific modifications. The core contribution comes down to the featurization of the time series, where the authors combine many previous methods and use multiple decompositions of the time series within the model. There are also some interesting architectural choices that enable the model's performance. The model performs well on downstream benchmarks, with the authors undergoing significant downstream testing.

**Strengths:**

- The writing is clear and straightforward, laying out the reasoning for the model components and describing them in sufficient detail. The architecture seems to have a lot of backing from literature, and the authors build nicely on previous work.
- It is encouraging to see that the combination of lots of disparate components of time series analysis results in a performative model. I think this is an important contribution to demonstrate, and the authors motivate the use of these features very well.
- The experiments in the paper are extensive and thorough, demonstrating the model’s applicability on a variety of downstream tasks. In addition, the appendix has a detailed number of experiments, which is helpful in observing model performance.

**Weaknesses:**

- One of the biggest weaknesses of the model is the resources required to run this architecture on practical time series data. A number of components of the model are very computationally expensive, including the full temporal attention and the internal FFT calls in the model. Some analysis of computational efficiency would be helpful.
- The performance, while better than baselines across most tasks, is only marginally better than other models. In the field of time series foundation models, where a plethora of models exist that all perform better or worse in a specific setting, this is not very impressive.
- There are no error bars anywhere in the results, making it difficult to determine significance of the observed gains in performance.
- There are a number of suspicious results from the ablations, including that the pretraining is not much better than training on specific datasets, indicating that the “generality” of the model might not be optimal. In the case where the pretraining does not help, the authors would need to test against task-specific models as the regime becomes different. In addition, the boost in model performance is marginal in the scaling law study, raising questions as to the generality of the model.

**Questions:**

- Have the authors tested the model on long-range datasets? What is the maximum length of the datasets used in this paper?
- Are the results over baselines statistically significant?

---

> ### Author Response · Authors · 2025-11-20
>
> We sincerely thank you for raising important comments and we are pleased to provide a detailed response below.
> 1. Computational Efficiency
> - GTM demonstrates competitive efficiency in both training and inference. As shown in Appendix B.3.8 (Table 24), GTM ranks second in parameter size, training speed and inference memory, and third in inference speed (0.165s/iter for batch size 128) and training memory, making it suitable for real-time deployment.
> - Additional experiments on two types of Nvidia GPUs (see Table 1) show that: for univariate data with 1440 input points and 96 prediction length, total latency is only 0.043s/item, with critical modules (FFT/iFFT and Fourier Attention) contributing minimally to overall computation time. Similar results also hold for multivariate data.
> - Table 1: Analysis of model inference latency and computational overhead in critical modules
> |GPU|Channels|Inference time(s/item)|FFT+iFFT(s/item)|Fourier Attention (s/item)|
> |-|-|-|-|-|
> |A100-40GB|1|0.043|0.0007|0.033|
> ||7|0.044|0.0007|0.034|
> ||862|0.142|0.0009|0.103|
> |RTX 4090 24GB|1|0.041|0.0007|0.031|
> ||7|0.041|0.0007|0.031|
> ||862|0.144|0.0009|0.107|
>
> 2. Marginal Performance Improvement Over Baselines
> - While average gains over SOTA baselines are modest due to strong existing results on open datasets, GTM achieves substantial improvements in specific scenarios. For example, on ETTh1 imputation with 50% masking, GTM improves MSE and MAE by 31.78% and 17.1% over the second-best models (GPT4TS and TimesNet, Appendix B.3, Table 16).
> - Beyond performance, another main contribution of GTM is its unified generative capability across forecasting, imputation, and anomaly detection tasks, without task-specific modifications—a flexibility lacking in previous TSFM models.
> - In future work, we plan to conduct more extensive experiments and validation on data from broader domains, especially by incorporating more pre-training data, with the goal of further improving TSFM performance.
>
> 3. Lack of Error Bars in experiments
> - Thank you for highlighting the absence of error bars in our experiments. In response, we add error bars by running 10 independent trials for long-term forecasting tasks, calculating the 95% confidence interval for all metrics (see Table 2) according to $Error\_bar=t*std$, where the t value is about 2.26 for 10 runs. Supplementary results for all experiments will be provided before the deadline.
> - Table 2 95% CI error bar for forecasting tasks
>
> | Dataset | Pred_len | MSE| MAE|
> |-|-|-|-|
> || 96 | 0.361±0.0029 | 0.399±0.0022 |
> |ETTh1| 192| 0.399±0.0031 | 0.424±0.0031 |
> || 336| 0.423±0.0032 | 0.439±0.0032 |
> || 720| 0.442±0.0108 | 0.464±0.0083 |
>
> 4. Questionable Generality and Pretraining Effectiveness
> - Pre-training consistently improves GTM’s performance across datasets and tasks (Tables 5, 6). While overall gains are modest—mainly because our baseline is already highly optimized—significant improvements appear in specific cases, such as a 12.4% MSE reduction for 720-step forecasting on ETTm1 and a 30.1% MAE reduction for weather imputation with 50% missing data (Appendix C.2). GTM also matches or outperforms SOTA models like UP2ME, with around 7% improvement.
> - GTM shows steady gains with increased pre-training data (e.g., 2.2% on ETTh1, 6.3% on Weather), comparable to SOTA models. For example, Time-MOE achieves only ~3% improvement when scaling data from 200B to 400B—an order of magnitude more than ours—while Timer shows ~4.6% gain on PEMS.
> - Overall, GTM’s design and pre-training strategy enable strong generality and flexible adaptation to various tasks, without task-specific modifications.
>
> 5. Evaluation on Long-Range Time Series Datasets
> - We provide detailed statistics for all datasets used in our experiments in Appendix B.2.1, Tables 10–13. Among these, the ETTm dataset represents the longest-range testing scenario, spanning over 725 days and containing up to 69,680 time points at a 15-minute sampling interval.
> - To further assess the scalability and generalization of GTM, we plan to extend our experiments to a broader range of publicly available long-range datasets across various domains in future work.
>
> 6. Statistical Significance of Results Over Baselines
> - GTM’s absolute improvements over SOTA baselines are sometimes limited, mainly because existing methods are already highly optimized. Nevertheless, GTM consistently matches or exceeds SOTA performance, and shows clear statistical advantages in best and second-best counts across tasks and datasets:
> - Forecasting (Table 15): GTM achieves 21 best and 22 second-best results out of 50 tasks, while the second best model only has 13 best and 15 second-best results.
> - Imputation (Table 16): GTM obtains 26 best and 12 second-best results out of 40 tests, compared to just 10 best results for the second best model.
> - Classification (Table 4): GTM achieves 5 best and 4 second-best results out of 10 tasks, while the second best model only has 3 best and 1 second-best.

---

> > ### Comment · Reviewer_bvHo · 2025-11-22
> >
> > 1. Thank you for this answer! I appreciate the analysis that points to the efficiency of each of the modules, this is helpful for understanding the efficiency of the method. However, this doesn’t answer how efficient this method would be on longer time series data. 96 prediction length is quite short, how would the efficiency scale to lengths that are 1 or 2 more orders of magnitude larger?
> > 2. Thank you for pointing out this experiment on ETTh1 masking. I think it would be helpful for authors to highlight some of these particular instances of larger gains in the main text for the final draft of the manuscript.
> > 3. These error bars are quite low, which is encouraging for the results. The authors should include error bars in the final draft of the manuscript to improve clarity.
> > 4. Thank you for taking the time to clarify these results. It is helpful to see comparisons to other pretrained time series models and the gain that GTM gets compared to those other methods. I think the results are still somewhat lackluster in this respect, but in context of other time series foundation models, these arguments make sense.
> > 5. Thank you for pointing to these statistics, this helps clarify the model’s performance on long-horizon tasks.
> > 6. I understand the performance gains of the model, which are indeed impressive. However, it is still not clear whether the performance gains are significant. The baselines being highly optimized does not excuse the lack of definitive evidence for GTM’s superiority as a model. I would want to see authors test on a non-standard, real-world dataset on which baselines perform very poorly. This would be more convincing as to the benefit of GTM.
> >
> > While I appreciate the authors’ response and effort, I cannot raise my score at this time. I believe the work is borderline at the moment, and I look forward to hearing input from the other reviewers.

---

> > > ### Author Response · Authors · 2025-11-25
> > >
> > > Thank you very much for your feedback on our rebuttal. We have provided further information and clarification on the points you raised, as follows:
> > >
> > > 1.How does model efficiency scale with significantly longer prediction lengths?
> > > - We further analyze model efficiency by conducting experiments with longer prediction lengths and larger input look-back windows. As shown in Table 1, the inference time remains nearly constant even as both the look-back window and prediction length increase by an order of magnitude. This illustrates that GTM does not fully saturate the computational resources of the A100 GPU, demonstrating high efficiency at the current scales and is well-suited for practical deployment in real-world sub-second streaming applications.
> > > -  From a model architecture perspective, GTM employs an autoregressive layer for output projection, rather than a conventional flatten + linear projection approach. It ensures that GTM's inference time mainly depends on the input look-back window length. When the output prediction length is less than or equal to the input window length, the inference time remains nearly constant.
> > > - Table 1. Model efficiency analysis for varying prediction lengths and look-back window lengths.
> > >
> > > | GPU       | Channels | Lookback Len. | Pred. Len. | Inference time (s/item) | FFT + iFFT (s/item) | Fourier Attention (s/item) |
> > > |-----------|----------|---------------|------------|------------------------|--------------------|---------------------------|
> > > | A100-40GB | 1        | 1440          | 96         | 0.043                  | 0.0007             | 0.033                     |
> > > |           | 1        | 2880          | 1440       | 0.043                  | 0.0007             | 0.033                     |
> > > |           | 1        | 5120          | 2880       | 0.043                  | 0.0007             | 0.033                     |
> > > |           | 1        | 14400         | 5120       | 0.043                  | 0.0007             | 0.033                     |
> > >
> > > 2.Highlighting instances of significant performance gains in the main text
> > > - Thank you for your suggestion. We'll highlight the instances where GTM achieves larger performance gains in the final draft of the manuscript.
> > >
> > > 3.Including error bars to improve clarity in the final draft of the manuscript.
> > > - We appreciate your feedback. We will include error bars in the final draft of the manuscript to enhance clarity after the completion of all experiments.
> > >
> > > 4.Comparative analysis with SOTA pretrained time series models
> > > - Thanks for your understanding.
> > >
> > > 5.Clarifying model performance on long-horizon forecasting tasks
> > > - Thanks for your confirmation, we'll add thses statistical information into the final draft of the manuscript.
> > >
> > > 6. Conduct more experiments on challenging, real-world datasets
> > > - Thank you for your suggestion. We have identified two particularly challenging real-world datasets that have not yet been fully optimized: one is an open PV solar energy forecasting dataset, and the other is the L2C (lead-to-cash) dataset, which combines observations of Business Key Performance Indicators (Biz-KPIs) and IT events. We are currently evaluating GTM and comparing it with reproduced SOTA baselines on these datasets. The results and corresponding analysis will be provided before the rebuttal deadline.
> > > - [1] Carreira Pedro et al., "A comprehensive dataset for the accelerated development and benchmarking of solar forecasting methods", https://zenodo.org/records/2826939
> > > - [2] L2C datasets, https://github.com/BizITObs/BizITObservabilityData/blob/main/README.md

---

> > > > ### Comment · Reviewer_bvHo · 2025-11-25
> > > >
> > > > 1. Thank you for providing this information. Why are the numbers constant per look-back length and prediction length? I suggest taking a closer look at this and maybe how you implemented the timing.
> > > > 2. Ok
> > > > 3. This sounds good, error bars are essential for the results.
> > > > 4. Ok
> > > > 5. Ok
> > > > 6. Thank you for evaluating the model on these real-world datasets; this should provide more information as the model’s performance.

---

> > > > > ### Author Response · Authors · 2025-11-27
> > > > >
> > > > > Thank for your further comments and suggestions. In response to your question regarding our test results that inference time is nearly constant across different prediction lengths with a fixed look-back window, we would like to offer further clarification on the design and implementation details as follows:
> > > > >
> > > > > There are generally three mainstream designs for projection layers in time series forecasting models. Below, we clarify these designs and discuss how they affect flexibility and inference efficiency:
> > > > >
> > > > > 1.Flatten layer with a linear projection (direct mapping)
> > > > > -  In this design, the backbone outputs a tensor of shape $[B, N_p, D]$ ($B$: batch size, $N_p$: number of patches, $D$: feature dimension). This tensor is flattened and mapped through a linear layer of shape $[N_p \times D, L]$, where $L$ is the prediction length. Models such as PatchTST (ICLR 2023), TimesNet(ICLR 23), Crossformer (ICLR 2021), and FreTS (NeurIPS 2023) follow this approach.
> > > > > - Limitations: The output head must be reconfigured for each $L$, limiting flexibility for variable-length forecasting. It is a clear disadvantage for TSFMs. Moreover, inference time increases with larger $L$, as the output head increases in size accordingly.
> > > > >
> > > > > 2.Autoregressive Approach
> > > > > - Strictly, the autoregressive approach is defined as the model predicts one future value at a time: at each step $t$, the model uses its previous prediction $\hat{y}_{t-1}$ (along with history) as input to predict $\hat{y}_t$. This process is repeated until the desired prediction length $L$ is reached.
> > > > > - Advantage: high flexibility—the same output head can generate variable-length sequences without retraining.
> > > > > - Limitations: inference time grows linearly with $L$ (since prediction is step-wise), and errors can accumulate as prediction length increases.
> > > > > - Consequently, SOTA TSFMs rarely adopt this approach for output projection.
> > > > >
> > > > > 3.Sequence to Sequence(seq2seq) aproach
> > > > > - In seq2seq, the model's projection layer is designed directly output entire prediction sequence of arbitrary length. In our implementation, we take backbone ourput representation with a shape $[B, N_p, D]$ as an input, to generate output time points $N_{pred} = N_p * patchsize$, which also corresponds to the look_back window length. During post-processing, the output is truncated to the target prediction length $L$.
> > > > > - Advantages: enables flexible output lengths, as the output head does not need to be specifically configured for each prediction length. This makes it well-suited for variable-length forecasting. Additionally, inference time is generally less sensitive to the output length compared to the autoregressive approach, since the entire sequence is generated simultaneously.
> > > > > - This explains why, in our tests (with a fixed look-back window), inference time remains almost constant for different prediction lengths less than or equal to the input window length.
> > > > > - SOTA TSFMs, such as TIMER(ICML 24), UP2ME(ICML 24), UniTS(NIPS 24) etc.,  apply this approach.
> > > > > - Note that the distinction between the Seq2Seq and autoregressive approaches can sometimes be ambiguous.  For example, although TIMER adopts a Seq2Seq framework in implementation, its paper refers to the output generation process as an autoregressive generation method.

---

### Official Review · Reviewer_B3pb · 2025-11-01

**Soundness:** 3
**Presentation:** 3
**Contribution:** 2
**Rating:** 4
**Confidence:** 4

**Summary:**

The paper proposes **GTM (General Time-series Model)**, a generative-task-agnostic time-series foundation model that enhances representation learning by integrating temporal and frequency-domain features. It introduces a Fourier attention mechanism to capture time-granularity-aware patterns and a unified pre-training framework (hybrid masking, 2D positional encoding, span shuffling) that unifies reconstruction and autoregressive objectives. GTM seamlessly adapts to diverse generative tasks (forecasting, imputation, anomaly detection) without task-specific modifications, outperforms SOTA models across benchmarks, and exhibits clear scaling behavior with model size and pre-training data.

**Strengths:**

1. This paper addresses a critical gap in existing time series foundation models by introducing a Fourier attention mechanism that explicitly captures time-granularity-aware patterns from the frequency domain.
2. As far as I am concerned, GTM is the first TSFMs that supports seamless adaptation to all generative tasks (forecasting, imputation, anomaly detection) without task-specific modifications (e.g., tokenization adjustments, projection header changes).
3. The paper conducts rigorous experiments across diverse benchmarks with fair and consistent enhancements, including 5 datasets for forecasting/imputation, 5 for anomaly detection, and 10 for classification.

**Weaknesses:**

1. The paper does not deeply explore the computational overhead of the Fourier attention mechanism—specifically, how FFT/iFFT operations (integrated into each decoder block) impact inference latency for real-time applications (e.g., streaming sensor monitoring).
2. The model analysis of this paper is insufficient. For example, no trade-off analysis (e.g., simplifying frequency modules to reduce latency) is provided, which is critical for practical deployment.
3. The TSFM baselines are outdated. Please include more additional baselines such as TabPFN, Sundial, TimeMOE.

**Questions:**

1. Comparing to the models listed in the baseline, how much more/less computational resource does GTM consume? Can GTM still remain high performance if the fourier transform calculation time is limited?
2. In the Fourier attention mechanism, different frequency patterns are taken into attention calculation by forming a vectorized feature. Does this strategy outperform mixing (by MLPs for instance) these frequency patterns into a representation embedding?

---

> ### Author Response · Authors · 2025-11-20
>
> Thank you for your valuable comments, which helped us further clarify and improve our work. Below, we address your concerns with additional experiments and explanations.
>
> 1.Analysis of the computational overhead of the Fourier Attention mechanism and Inference Latency
>
> - GTM incorporates Fourier Attention across all backbone layers, boosting performance while remaining efficient for real-time use. We have compared to four baselines (Time-MOE, GPT-2(6)-768, TimesNet-768, FEDformer-768) in Table 24, Appendix B.3.8. The results show that GTM ranks second in parameter size, training speed and inference memory, and third in inference speed (0.165s/iter for batch size 128) and training memory, demonstrating suitability for sub-second real-time industrial applications.
>
> - We additionally analyze the computational overhead of Fourier Attention and FFT/iFFT. On both A100 and RTX 4090 GPUs,  as shown in Table1, GTM achieves fast inference: for univariate data with 1440 input points and 96 prediction length, total latency is only 0.043s/item, with FFT/iFFT and Fourier Attention contributing minimal overhead. Similar results hold for multivariate data, demonstrating GTM’s low-latency, real-time capability.
>
> - Table 1: Analysis of model inference latency and computational overhead in critical modules
> |GPU|Channels|Inference time(s/item)|FFT+iFFT(s/item)|Fourier Attention (s/item)|
> |-|-|-|-|-|
> |A100-40GB|1|0.043|0.0007|0.033|
> ||7|0.044|0.0007|0.034|
> ||862|0.142|0.0009|0.103|
> |RTX 4090 24GB|1|0.041|0.0007|0.031|
> ||7|0.041|0.0007|0.031|
> ||862|0.144|0.0009|0.107|
>
> 2.Trade-off analysis (simplifying frequency modules to reduce latency)
> - We evaluated how increasing the number of low-rank modules in Fourier Attention affects inference latency(see Table 2, using the same settings as Table 1). More modules lead to a gradual, sub-linear increase in processing time, but even with 20 modules, latency stays below 0.1s/item, meeting real-time requirements. This allows GTM to flexibly balance model expressiveness and speed. For future work, we plan to extend Fourier Attention to a MoE architecture and explore distributed, parallel processing to further reduce latency.
> - Table 2 Trade-off analysis
>
> |Low-rank modules |Channels |Inference time(s/item) |Fourier Attention(s/item) |
> |-|-|-|-|
> |1|1|0.030|0.020|
> ||7| 0.030| 0.020|
> |5|1| 0.043| 0.033|
> ||7| 0.044| 0.034|
> |10 |1| 0.060| 0.049|
> ||7 | 0.061| 0.050|
> |15 |1| 0.076| 0.065|
> ||7| 0.078| 0.066|
> |20|1| 0.092|0.080|
> ||7| 0.094| 0.081|
>
>
> 3.Additional baseline model comparison
> - To update our baseline comparisons in forecasting tasks, we included Sundial-small and Time-MOE-base, both of which are similar in size to GTM and were tested under comparable conditions (see Table 3). GTM achieved the best results on 15 tasks, compared to 14 for Time-MOE-base when evaluated on the same datasets. In contrast, Sundial-small achieved the best result on only one task, further highlighting GTM’s competitiveness and robustness.
> - Although TabPFN is primarily designed for tabular data, it can be adapted for time series prediction. However, direct comparison is challenging due to differences in datasets and evaluation settings. We plan to reproduce TabPFN and conduct a fair comparison in future work under consistent conditions.
> - Table 3 Additional performance comparison with SOTA models in forecasting tasks
>
> |Models||**GTM**||Time-MOE||Sundial||
> |-|-|-|-|-|-|-|-|
> |dataset|pred_len|MSE|MAE|MSE|MAE|MSE|MAE|
> ||96|0.360|0.398|0.345|**0.373**|**0.341**|0.381|
> ||192|0.397|0.422|**0.372**|**0.396**|0.381|0.408|
> |ETTh1|336|0.420|0.437|**0.389**|**0.412**|0.405|0.424|
> ||720|0.438|0.457|**0.410**|**0.443**|0.433|0.458|
> ||AVG|0.404|0.429|**0.379**|**0.406**|0.390|0.418|
> ||96|**0.282**|0.341|0.286|**0.334**|0.292|0.342|
> ||192|0.325|0.366|**0.307**|**0.358**|0.337|0.376|
> |ETTm1|336|**0.353**|**0.385**|0.354|0.390|0.370|0.401|
> ||720|**0.396**|**0.410**|0.433|0.445|0.418|0.433|
> ||AVG|**0.339**|**0.376**|0.345|0.381|0.354|0.388|
> ||96|**0.147**|**0.197**|0.151|0.203|0.158|0.206|
> ||192|**0.192**|**0.241**|0.195|0.246|0.205|0.253|
> |weather|336|0.250|0.291|**0.247**|**0.288**|0.254|0.290|
> ||720|**0.310**|**0.334**|0.352|0.366|0.315|0.336|
> ||AVG|**0.225**|**0.266**|0.236|0.275|0.233|0.271|
> ||96|**0.131**|**0.225**|-|-|0.134|0.231|
> ||192|**0.149**|**0.243**|-|-|0.154|0.251|
> |Electricity|336|**0.166**|**0.259**|-|-|0.174|0.271|
> ||720|**0.201**|**0.292**|-|-|0.215|0.307|
> ||AVG|**0.161**|**0.254**|-|-|0.169|0.265|
> |Best Count||13(8)|12(7)|6|8|1|0|
>
> 4.Fourier Attention vs. MLP-based Mixing for Representation Learning
> - MLPs mainly capture local patterns and blend frequencies, while Fourier Attention models global, complex dependencies across frequencies. Since time series often show distinct patterns at different granularities, Fourier Attention enables flexible, interpretable representation learning. We chose Fourier Attention for its superior ability to capture multi-granularities relationships in large-scale time series data.

---

### Author Response · Authors · 2025-12-01
**General response to all the comments and suggestions.**

Dear reviewers,

We would like to express our sincere gratitude to all reviewers for your thoughtful and constructive feedback on our manuscript. Your insightful comments and suggestions have been invaluable in helping us clarify and strengthen our paper. We have carefully addressed each point raised and have updated the submitted PDF file accordingly.

In this revision, we have focused on two major common concerns highlighted during the review process:

1.Hybrid masking mechanism:
- We have added a detailed explanation of the hybrid masking strategy and its control mechanism in Section 3.3, with implementation details in Appendix B.2.3. This clarifies the design rationale, operation, and hyperparameter settings, ensuring transparency and reproducibility.

2.Model efficiency and inference time analysis:
- We have introduced a new Section 4.9 in the main text, which includes:
    - A comparison of model parameters and efficiency with SOTA TSFMs.
    - A computational overhead analysis of critical modules (FFT/iFFT and Fourier Attention)
    - A trade-off analysis of inference latency across different frequency modules.
- The results demonstrate that GTM is not only competitive for deployment but also efficient enough for real-time, sub-second streaming applications.

In addition, we have conducted further experiments and expanded our descriptions in response to reviewer suggestions:
- Appendix B.2.1:
    - We now highlight that the ETTm dataset represents the longest-range testing scenario in our experiments, spanning over 725 days and containing up to 69,680 time points sampled at 15-minute intervals.
- Appendix B.3.1:
    - We performed error bar analysis with 95% confidence intervals by running 10 independent trials across different datasets. We updated the calculation follows a more rigorous formulation: $errorbar = t_{0.025,\,n-1} \times \frac{std}{\sqrt{n}}$. The consistently low error bars for both MSE and MAE indicate high reliability and stability of our results.
    - We extended baseline comparisons to include two recent TSFMs: Sundial-Small and Time-MOE-Base. GTM outperforms Sundial-Small and shows slightly better results than Time-MOE-Base, confirming its robustness across diverse datasets and prediction horizons.
    - To address concerns about dataset over-exploitation, we introduced two challenging real-world datasets: an open PV (Photovoltaic) solar energy forecasting dataset and the L2C (lead-to-cash) dataset, which integrates business KPIs with IT event logs. Comparisons with reproduced SOTA models (PatchTST and TimesNet) show that GTM consistently achieves SOTA performance on both new datasets.
        - The most significant improvement is observed on the L2C dataset with a prediction length of 720.
            - For MSE, GTM achieves a score of 0.7170, outperforming the second-best method (TimesNet, 1.2984) which means a 44.8% reduction.
            - For MAE, GTM attains a value of 0.5218 compared to PatchTST’s 0.8460, resulting in a 38.3% reduction.
- Appendix B.3.8
    - We further analyzed model efficiency under longer prediction lengths and larger look-back windows. Results confirm that GTM does not saturate A100 GPU resources, demonstrating scalability and suitability for real-time streaming applications.
    - We also provided an architectural discussion of three mainstream output projection designs in time series forecasting, explaining why GTM’s seq2seq approach maintains nearly constant inference latency across varying prediction lengths.

All mentioned updates, additional experiments, and enhanced methodological explanations have been incorporated into the revised manuscript. We believe these revisions have addressed all concerns raised during the review and rebuttal period.

We deeply appreciate the reviewers’ time and effort in helping us improve this work. Thank you for your valuable contributions to strengthening our paper.

---

### Meta-Review · Area_Chair_37V9 · 2026-01-02

**Summary:**

Reviewers generally agree the manuscript is clearly written and the experimental coverage is unusually extensive across tasks and datasets, with one reviewer finding the empirical motivation for granularity-dependent frequency structure and the resulting architectural design particularly strong and aligned with the foundation-model goal. While some reviewers view the gains over competitive baselines as modest on average, the method appears consistently competitive and occasionally substantially better in specific regimes, and the single model / no task-specific modifications aspect is a meaningful practical contribution in an increasingly fragmented TSFM space.

Importantly, the rebuttal directly addressed the largest acceptance blockers raised by multiple reviewers: it adds concrete efficiency and module-overhead measurements (including FFT/iFFT and Fourier attention contributions), provides a trade-off analysis for the frequency module, expands comparisons to more recent TSFM baselines (e.g., Sundial/Time-MOE and discussion of UniTS/Moirai settings), and begins to supply statistical evidence via repeated runs and confidence intervals (at least on forecasting). The remaining concerns, fully convincing statistical significance across all tasks, tighter validation of long-horizon scaling methodology, and more decisive differentiation on challenging real-world datasets, are legitimate, but they read as strengthening evidence and presentation rather than fundamental flaws in the technical direction.

**Reviewer Concerns:**

I believe that the main reviewer concerns were substantially addressed. First, computational efficiency and overhead, raised by B3pb, bvHo, and boeL, were clarified with detailed latency, memory, and throughput analyses, including module-level breakdowns for FFT/iFFT and Fourier attention and trade-off studies varying the number of low-rank modules; these results indicate GTM is feasible for real-time settings. Second, baseline coverage was expanded and clarified: the authors added comparisons to Sundial and Time-MOE, and carefully positioned results against UniTS and Moirai under matched settings, addressing concerns about outdated or incomplete baselines (B3pb, boeL). Third, underspecified methodological details were fixed, notably the hybrid masking control (pred_ratio), directly resolving mmJe's concern. Fourth, statistical reliability concerns were partially mitigated by adding repeated runs with confidence intervals for forecasting tasks (bvHo). Finally, questions about long-horizon applicability were addressed with additional timing experiments and clarification of the projection design explaining why inference time is largely insensitive to prediction length.

**Reviewer Scores:**

At least two reviewers would raise their rating if they had been able to participate fully in the discussion, because the main concerns overlap and were effectively addressed by the authors.

---

### Decision · Program_Chairs · 2026-01-26

Accept (Poster)